EMBO
Molecular Medicine

# Epigenetic silencing of SALL2 confers tamoxifen resistance in breast cancer

Liping Ye[1,†] iD, Chuyong Lin[1,†], Xi Wang[1,†], Qiji Li[2,†] iD, Yue Li[1], Meng Wang[1], Zekun Zhao[3], Xianqiu Wu[4], Dongni Shi[1], Yunyun Xiao[1], Liangliang Ren[5], Yunting Jian[1], Meisongzhu Yang[5,6], Ruizhang Ou[7], Guangzheng Deng[1], Ying Ouyang[1], Xiangfu Chen[1], Jun Li[5,*] iD & Libing Song[1,**] iD

## Abstract

Resistance to tamoxifen is a clinically major challenge in breast cancer treatment. Although downregulation of estrogen receptor-alpha (ERα) is the dominant mechanism of tamoxifen resistance, the reason for ERα decrease during tamoxifen therapy remains elusive. Herein, we reported that Spalt-like transcription factor 2 (SALL2) expression was significantly reduced during tamoxifen therapy through transcription profiling analysis of 9 paired primary pre-tamoxifen-treated and relapsed tamoxifen-resistant breast cancer tissues. SALL2 transcriptionally upregulated ESR1 and PTEN through directly binding to the DNA promoters. By contrast, silencing SALL2 induced downregulation of ERα and PTEN and activated the Akt/mTOR signaling, resulting in estrogen-independent growth and tamoxifen resistance in ERα-positive breast cancer. Furthermore, hypermethylation of SALL2 promoter was found in tamoxifen-resistant breast cancer. Importantly, *in vivo* experiments showed that DNA methyltransferase inhibitor-mediated SALL2 restoration resensitized tamoxifen-resistant breast cancer to tamoxifen therapy. These findings shed light on the mechanism of SALL2 in regulation of ER and represent a potential clinical signature that can be used to categorize breast cancer patients who may benefit from co-therapy with tamoxifen and DNMT inhibitor.

**Keywords** breast cancer; ESR1; methylation; SALL2; tamoxifen resistance
**Subject Categories** Cancer; Chromatin, Transcription & Genomics

## Introduction

Endocrine therapies, including selective estrogen receptor modulators (SERMs), aromatase inhibitors (AIs), and selective estrogen receptor downregulators (SERDs), are the mainstay of treatments for estrogen receptor-alpha (ERα)-positive (ER[+]) breast cancer, which accounts for approximately 70% of all breast cancers (Musgrove & Sutherland, 2009). Tamoxifen, which competes with estrogen for ERα and inhibits the stimulatory effect of estrogen for tumor growth, is the first-line therapy and most widely used endocrine agent for adjuvant therapy, particularly in premenopausal women (Swaby *et al*, 2007). Although tamoxifen therapy has a beneficial effect on survival time and reduces recurrent events in breast cancer patients (Cuzick *et al*, 2007), approximately 30–40% of patients with breast cancer receiving adjuvant tamoxifen therapy still relapse or progress to deadly advanced metastatic stages within 15 years of follow-up (EBCTC, 1998; EBCTC *et al*, 2011). The mortality rate of ER[+] breast cancer patients is largely attributed to the development of tamoxifen-resistant recurrent tumors. However, the precise mechanism that leads to tamoxifen resistance during tamoxifen therapy remains obscure.

Although several mechanisms have been proposed for tamoxifen resistance, including modulation of estrogen receptor signaling, activation of alternative oncogenic signaling such as PI3K/Akt/mTOR pathway and growth factor signaling, and deregulation of oncogenic proteins and transcription factors, dysregulation of ER and its pathway remain central to endocrine resistance (Tryfonidis *et al*, 2016). It is well established that ER expression level is a determinant of tamoxifen response in ER[+] breast cancer, and loss of ER expression is a crucial factor contributing to tamoxifen resistance (Musgrove & Sutherland, 2009; Osborne & Schiff, 2011). Notably, the transcriptional levels of ESR1 are significantly decreased in relapsed lesions compared with primary tissues in tamoxifen-treated breast cancer

1   State Key Laboratory of Oncology in South China, Collaborative Innovation Center for Cancer Medicine, Sun Yat-sen University Cancer Center, Guangzhou, China
2   Department of Orthopaedic Surgery, The Seventh Affiliated Hospital, Sun Yat-sen University, Shenzhen, China
3   Division of Biosciences, University College London, London, UK
4   Clinical Experimental Center, Department of Pathology (Clinical Biobanks), Jiangmen Central Hospital, Affiliated Jiangmen Hospital of Sun Yat-sen University, Jiangmen, Guangdong, China
5   Department of Biochemistry, Zhongshan School of Medicine, Sun Yat-sen University, Guangzhou, China
6   Key Laboratory of Protein Modification and Degradation, School of Basic Medical Sciences, Affiliated Cancer Hospital & Institute of Guangzhou Medical University, Guangzhou, China
7   Department of Pathology, School of Basic Medical Science, Southern Medical University, Guangzhou, China
    *Corresponding author. Tel: +86 20 87335828; Fax: +86 20 87335828; E-mail: lijun37@mail.sysu.edu.cn
    **Corresponding author. Tel: +86 20 87343187; Fax: +86 20 87335828; E-mail: songlb@sysucc.org.cn
    †These authors contributed equally to this work

(Johnston *et al*, 1995; Drury *et al*, 2011; Kim *et al*, 2011), suggesting that ESR1 might be transcriptionally repressed during acquired resistance to tamoxifen. Though multiple regulators, such as Notch3 (Dou *et al*, 2017), NR2E3 (Park *et al*, 2012), and MEL-18 (Lee *et al*, 2015), have been reported to be involved in tamoxifen resistance using cell or mouse models, the real contribution of these factors to tamoxifen resistance during therapy remains unclear.

Spalt-like transcription factor 2 (SALL2), a member of the C2H2 zinc finger transcription factor family, plays vital roles in normal development and is implicated in disease progression (de Celis & Barrio, 2009; Kelberman *et al*, 2014). Recently, SALL2 was found to be downregulated in several human cancer types, including breast cancer, ovarian cancer, and esophageal squamous cell carcinoma (Liu *et al*, 2007; Sung *et al*, 2013; Luo *et al*, 2017). Loss of SALL2 could abrogate serum deprivation-induced cell cycle arrest, but SALL2 overexpression suppressed cell growth (Liu *et al*, 2007). These studies suggest that SALL2 may be acting as a tumor suppressor. However, Suva *et al* (2014) reported that SALL2 may also function as an oncogenic protein by converting differentiated glioblastoma cells into stem-like tumor-propagating cells, resulting in glioblastoma propagation. These results suggest that the functional roles of SALL2 may be dependent on cell type and on potential signaling partners available in the cellular environment.

Herein, we found that SALL2 was significantly downregulated during tamoxifen therapy, and loss of SALL2 conferred estrogen-independent growth and tamoxifen-resistant phenotype in ER$^+$ cancer cells by decreasing ER$\alpha$ and PTEN expression. *In vivo* experiments revealed that restoration of SALL2 using DNA methyltransferase (DNMT) inhibitor resensitized tamoxifen-resistant breast cancer to tamoxifen therapy. These results uncover the crucial role of SALL2 in modulation of tamoxifen response and identify a subset of breast cancer patients who could benefit from co-therapy with tamoxifen and DNMT inhibitor.

# Results

## SALL2 expression correlates with response of tamoxifen therapy in breast cancer

To determine the clinically relevant mechanism underlying tamoxifen resistance, RNA-sequencing (RNA-seq) analysis was performed on 9 paired breast cancer samples, comprising primary breast cancer tissues from the same individuals at diagnosis and at relapse after tamoxifen treatment (Fig 1A). The patients' details are provided in Appendix Table S1. RNA-seq analysis revealed that a total of 196 genes, including 155 downregulated genes and 41 upregulated genes, were dysregulated in the relapsed tamoxifen-resistant breast cancer tissues compared to the primary tissues (Fig 1B). We found that ESR1 mRNA levels were downregulated in eight out of nine relapsed lesions compared to their corresponding primary tumors (Figs 1B and EV1A and B), which provided additional evidence that ESR1 was transcriptionally repressed during tamoxifen therapy. We further screened the potential transcription factors that may regulate ESR1, and found that among 196 dysregulated genes, the expression of 50 genes was significantly correlated with ESR1 ($r > 0.6$), and only two transcription factors, SALL2 and NKX3-1, were part of this dataset (Fig 1C and D).

Next, we validated the role of SALL2 and NKX3-1 in resistance to tamoxifen using gene-silencing experiments. As shown in Figs 1E and EV1C–F, silencing SALL2 significantly decreased ESR1 transcription levels and abolished tamoxifen-induced growth inhibition in ER$^+$ breast cancer cells, including MCF7, T47D, and ZR-75-1. However, silencing of NKX3-1 did not trigger the same effects. These results indicate that loss of SALL2 might contribute to ESR1 downregulation and tamoxifen resistance. Additionally, analysis of Kaplan–Meier plotter database (Gyorffy *et al*, 2010) indicated that lower SALL2 expression significantly correlated with poorer relapse-free survival (RFS), distant metastasis-free survival (DMFS), and overall survival (OS) in all breast cancer cases and in cases of ER$^+$ breast cancer with tamoxifen therapy, but not in ER$^-$ breast cancer cases (Fig 1F; Appendix Fig S1A and B). No correlation was observed between NKX3-1 expression and clinical outcomes in ER$^+$ breast cancer cases (Fig 1F). Of importance, we found that SALL2 expression was significantly decreased in relapsed metastatic tumors compared to the paired primary tumors (Fig 1G). Moreover, two alternative splice transcripts of SALL2, E1 and E1A, have been identified previously (Hermosilla *et al*, 2017). Analysis of the collected breast cancer tissues and of a TCGA breast cancer dataset revealed that only the E1A isoform of SALL2 was expressed in breast cancer samples (Appendix Fig S2A and B). Taken together, these results suggest that SALL2 expression correlates with response to tamoxifen therapy in breast cancer.

## SALL2 downregulation is associated with poor outcomes in human breast cancer

To further determine the role of SALL2 in the response to tamoxifen treatment, the expression of SALL2 was assessed in 238 clinical breast cancer specimens, including 90 cases of tamoxifen-treated breast cancer (Appendix Table S2). SALL2 expression was quantified via immunohistochemistry (IHC) staining using specific anti-SALL2 antibody (Appendix Fig S3A). IHC analysis revealed that SALL2 protein was detectable in normal breast tissues and mainly localized in the nuclei and to a lesser extent in the cytoplasm (Appendix Fig S3B). Previous studies have reported that serum deprivation induces SALL2 expression (Liu *et al*, 2007) and that highly proliferative cancer cells usually create local hypoxia and nutrient deficiencies (Hanahan & Weinberg Robert, 2011; Katheder *et al*, 2017). As expected, SALL2 was markedly expressed in tamoxifen-sensitive ER$^+$ breast cancer tissues (Appendix Fig S3B and Fig 2A). However, tamoxifen-resistant ER$^+$ breast cancer tissues and ER$^-$ breast cancer tissues showed further decreased SALL2 expression (Appendix Fig S3B and Fig 2A). Statistical analysis revealed that SALL2 expression was positively correlated with ER$\alpha$ level (Fig 2B). Consistently, SALL2 expression in ER-positive breast cancer cell lines, including MCF7, T47D, ZR-75-1, MDA-MB-361, MDA-MB-415, and ZR-75-30, was significantly higher than that in ER-negative breast cancer cell lines (BT-20, BT-549, and MDA-MB-231) and in normal immortalized mammary epithelial cell lines MCF10A and MCF12A, which are also ER-negative (Appendix Fig S3C). Meanwhile, we found that breast cancer patients with ER$^+$ status or higher SALL2 expression had longer disease-free survival (DFS) and OS compared to breast cancer patients with ER$^-$ status or lower SALL2 expression (Fig 2C and D; Appendix Table S3).

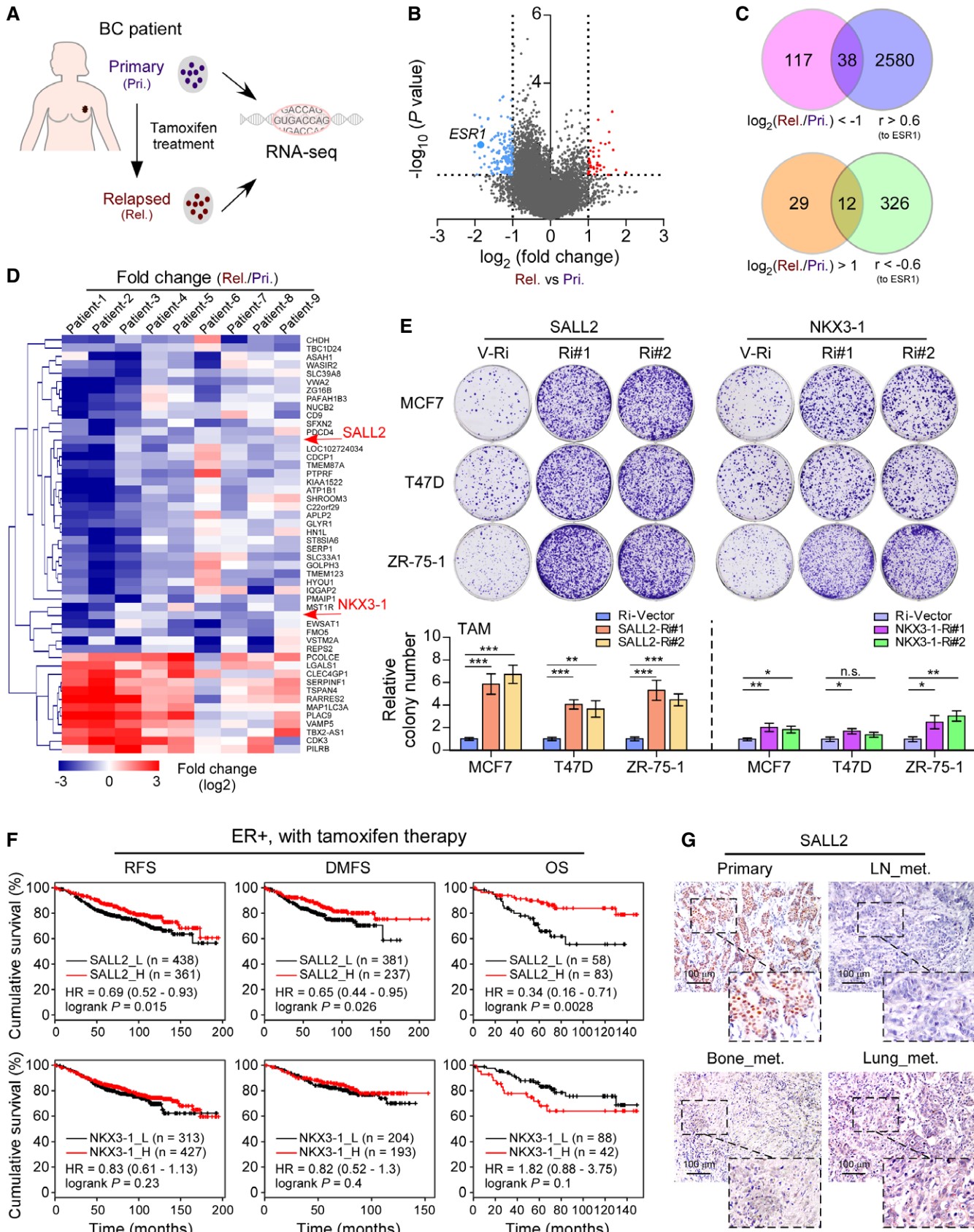

Figure 1.

◄

**Figure 1. SALL2 reduction contributes to tamoxifen resistance in breast cancer.**

A   Schematic diagram of the breast cancer tissues, comprising 9 pre-tamoxifen-treated primary breast cancer tissues and paired relapsed tamoxifen-resistant tissues from the same individuals, used for RNA-seq analysis. Primary, Pri.; relapsed, Rel.

B   Volcano plot for gene expression in the RNA-seq analysis by comparing the paired relapsed tamoxifen-resistant breast cancer tissues and pre-tamoxifen-treated primary breast cancer tissues from the same individuals. The blue dots represent the 155 downregulated genes and red dots represent the 41 upregulated genes among 196 genes dysregulated in the relapsed tamoxifen-resistant breast cancer tissues compared to the primary breast cancer tissues analyzed by RNA-seq.

C   Venn diagram, generated with 196 significantly changed genes comparing relapsed tamoxifen-resistant breast cancer tissues with pre-tamoxifen-treated tissues ($P < 0.05$), showing that expression of 50 genes, including 38 downregulated and 12 upregulated genes, was correlated with ESR1 expression ($P < 0.05$).

D   Heatmap of RNA-seq gene expression fold change (Rel./Pri.) from the 50 dysregulated genes in Venn diagram (C). Each column is denoted as one patient (patient 1 to patient 9).

E   Representative images (upper panel) and quantification (lower panel) of colony formed by three ER$^+$ breast cancer cell lines (MCF7, T47D, and ZR-75-1) expressing controls and shRNAs for SALL2 or NKX3-1 after tamoxifen (TAM) treatment for 24 h.

F   Kaplan–Meier (KM) plotter analysis of correlation of SALL2 (upper panel) and NKX3-1 (lower panel) expression with RFS, DMFS, and OS of patients with ER$^+$ breast cancer with tamoxifen therapy. The data were assessed using the Kaplan–Meier method with the log-rank test (auto-select best cutoff).

G   Representative IHC images of SALL2 protein in primary and metastatic breast cancer tissues. Scale bars: 100 μm.

Data information: In (E), data are presented as mean ± SD, and P-values were determined by one-way ANOVA test, $n = 3$. *$P < 0.05$, **$P < 0.01$, ***$P < 0.001$, n.s., no significance. Exact P-values are specified in Appendix Table S10.
Source data are available online for this figure.

Accordingly, patients with lower SALL2 expression and ER$^-$ status showed poorer DFS and OS compared with other groups of patients with breast cancer (Fig 2E). SALL2 expression was significantly lower in the tamoxifen-treated ER$^+$ breast tumors with relapse or metastasis than that in the primary tumors without relapse or metastasis (Fig 2F; Appendix Table S4).

We then further tested whether SALL2 expression could be used to identify breast cancer patients that might benefit from tamoxifen therapy. As shown in Fig 2G, the positive correlation between SALL2 expression and better prognosis was also significant in tamoxifen-treated ER$^+$ breast cancer patients. In addition, multivariate Cox regression analysis revealed that lower SALL2 expression was an independent prognostic factor for worse DFS and OS in all breast cancer cases and in ER$^+$ cases (Appendix Tables S5 and S6). Taken together, these results indicate that lower SALL2 could potentially predict higher risk of metastatic relapse and poorer prognosis in ER$^+$ breast cancer patients treated with tamoxifen therapy.

**Silencing SALL2 downregulates the transcription of ESR1**

To investigate the mechanism by which SALL2 transcriptionally regulates ESR1, we established a tamoxifen-resistant MCF7-TMR cell line, derived from tamoxifen-sensitive ER$^+$ breast cancer MCF7 cell line (Fig EV2A). The mRNA and protein levels of SALL2 and ESR1/ERα were significantly decreased in MCF7-TMR cells compared to parental MCF7 cells (Fig EV2B and C). Interestingly, we found that expression of ESR1 and ERα was significantly decreased in SALL2-silenced MCF7 cells but increased in SALL2-overexpressing

MCF7-TMR and ZR-75-30 cells (Fig 3A and B). These results indicate that ESR1 might be a downstream target of the SALL2 gene. Consistent with this hypothesis, we found a putative SALL2-binding site (GGGTGGG) on the promoter region of ESR1. Luciferase reporter assays revealed that silencing SALL2 significantly reduced the reporter activity of the ESR1 promoter, but had no effect on the mutant ESR1 promoter that contained mutant SALL2-binding site (Fig 3C). Furthermore, endogenous SALL2 could bind the putative binding site on the ESR1 promoter region, while silencing SALL2 significantly decreased the enrichment of p300, RNA polymerase II, and H3K4me3 on ESR1 promoter, as assessed by chromatin immunoprecipitation (ChIP) analyses (Fig 3D and E). Taken together, these results demonstrate that SALL2 transcriptionally activates ESR1 via directly interacting with ESR1 promoter in breast cancer cells.

To explore whether SALL2 regulates ERα activity, an estrogen-responsive element (ERE)-luciferase reporter was created. As shown in Fig 3F, overexpressing SALL2 significantly increased ERα transcriptional activity upon E2 treatment, which was then decreased upon tamoxifen treatment. Consistently, expression of TFF1 (trefoil factor 1) and PGR (progesterone receptor), which are classical ERα target genes, was significantly decreased in SALL2-silenced MCF7 cells and increased in SALL2-overexpressing MCF7-TMR and ZR-75-30 cells in response to estrogen treatment (Fig 3G). However, tamoxifen treatment inhibited the expression of these genes in SALL2-overexpressed cells, whereas it had no effect in SALL2-silenced MCF7 cells (Fig 3G). These results suggest that SALL2 functionally regulates the ER-dependent transcriptional network.

**Figure 2. SALL2 downregulation correlates with poorer outcomes of breast cancer.**

A   Representative IHC images of SALL2 and ERα in tamoxifen-treated ER$^+$ breast cancer without or with relapse and in ER$^-$ breast cancer tissues. Scale bars: 100 μm.

B   Bar graph showing positive correlation between SALL2 level and ERα expression status.

C–E   Kaplan–Meier analysis of DFS (upper panel) and OS (lower panel) curves for patients with ER$^-$ breast cancer or ER$^+$ breast cancer (C), with low SALL2 expression (SALL2_L) or high SALL2 expression (SALL2_H) (D), and with ER$^-$ SALL2_L or ER$^+$ SALL2_H, or other breast cancer (E).

F   Bar graph showing the proportion of SALL2 expression in ER$^+$ breast cancer tissues with relapse/metastasis (Rep./Met.) or without relapse/metastasis (No Rep./Met.).

G   Kaplan–Meier analysis of DFS (left panel) or OS (right panel) curves for patients with tamoxifen-treated ER$^+$ breast cancer with low SALL2 expression versus high SALL2 expression.

Data information: In (B and F), P-values were determined by $\chi^2$ test, $n = 238$ (B), $n = 90$ (F). In (C–E), P-values were determined by log-rank test, $n = 238$. In (G), P-values were determined by log-rank test, $n = 90$.

►

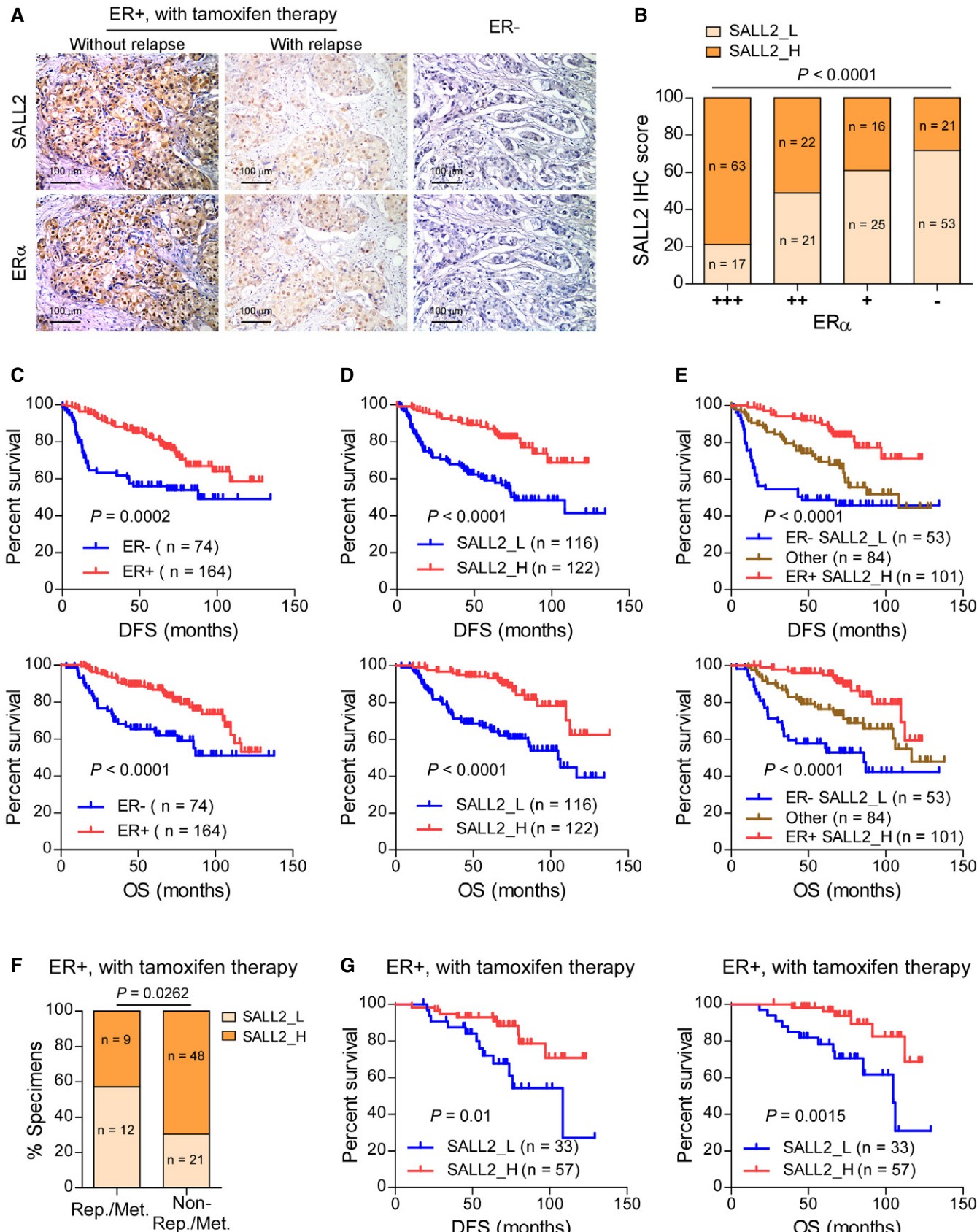

Figure 2.

## Silencing SALL2 induces tamoxifen resistance by downregulating levels of ER

We further investigated whether ER downregulation was required for SALL2 silencing-induced tamoxifen-resistant phenotype. As shown in Fig EV3A, silencing SALL2 decreased the sensitivity of MCF7 cells to tamoxifen treatment, whereas overexpressing SALL2 had the opposite effect. SALL2-silenced MCF7 cells treated with or without tamoxifen displayed a high percentage of cells in S phase, while SALL2-overexpressing MCF7 cells exhibited increased proportion of G1/S phase-arrested cells upon tamoxifen treatment (Fig EV3B). Furthermore, silencing SALL2 increased the growth rate of MCF7 cells independently of E2 or tamoxifen treatment, while restoration of ER expression abolished the estrogen-independent and tamoxifen-resistant phenotypes of SALL2-silenced MCF7 cells (Fig EV3C). By contrast, SALL2 overexpression-induced growth inhibition of MCF7-TMR and ZR-75-30 cells was abolished by E2 treatment, while silencing ESR1 dramatically reduced SALL2-induced increasing response to tamoxifen treatment (Fig EV3C). Similarly, depletion of ESR1 significantly inhibited the pro-apoptotic effect of tamoxifen on SALL2-overexpressing MCF7-TMR and ZR-75-30 cells (Fig EV3D). Altogether, these results indicate that ER loss is required for SALL2 silencing-induced estrogen-independent growth and tamoxifen-resistant phenotype of ER-positive cancer cells.

We further evaluated the effect of SALL2 loss on estrogen-independent and tamoxifen-resistant phenotype in breast cancer cells *in vivo*. MCF7/SALL2-Ri cells or control cells were injected into the mammary fat pads of female mice 1 week after implantation of E2 pellets, and tamoxifen treatment started once the tumors had reached a volume of approximately 200 mm³ (Fig 4A). As shown in Fig 4A and B, silencing SALL2 in the MCF7 xenografts significantly decreased the anti-tumor effect of tamoxifen treatment. Consistently, ERα expression was drastically reduced in SALL2-silenced tumor cells (Fig 4C). Strikingly, we found that MCF7/SALL2-Ri#1 cells gave rise to tumors in the mammary fat pad of nude mice after 1 month of inoculation without estrogen pellet implantation, whereas MCF7/control cells failed to do so (Fig 4D and E). Consistently, ERα expression was barely detected in SALL2-silenced tumors without E2, but strongly expressed in E2-treated MCF7/

control xenografts (Fig 4E), which further supports the hypothesis that silencing SALL2 confers estrogen-independent tumorigenicity to ER⁺ breast cancer cells.

## Restoring SALL2 expression increases sensitivity to tamoxifen in tamoxifen-resistant breast cancer

To determine whether upregulation of SALL2 could improve the anti-tumor effect of tamoxifen, a MCF7-TMR cell line with a doxycycline-inducible SALL2 cDNA (SALL2-Dox) was generated. As shown in Fig 4F and G, doxycycline treatment not only induced expression of SALL2 and ERα in MCF7-TMR/SALL2-Dox cells, but also resulted in significant upregulation of multiple ER downstream target genes, including CA12, RET, STC2, KAAG1, PMEPA1, MAST4, MSX2, and GFRA1. Moreover, doxycycline-treated MCF7-TMR/SALL2-Dox cells were more sensitive to tamoxifen than control cells (Fig 4H). Importantly, this *in vivo* model provided evidence that doxycycline treatment significantly enhanced the anti-tumor effect of tamoxifen on MCF7-TMR/SALL2-Dox tumors compared with control tumors (Fig 4I). These results further confirm that restoring SALL2 increases sensitivity to tamoxifen in tamoxifen-resistant breast tumors.

## Silencing SALL2 activates Akt/mTOR signaling via downregulation of PTEN

Gene Set Enrichment Analysis (GSEA) of the Cancer Genome Atlas (TCGA) Breast invasive carcinoma (BRCA) dataset indicated that low expression of SALL2 correlates with a PI3K/Akt/mTOR signaling signature (Fig 5A; Appendix Fig S4A), suggesting that loss of SALL2 might activate PI3K/Akt/mTOR pathway. Consistent with this hypothesis, silencing SALL2 in MCF7 cells dramatically increased the phosphorylation levels of Akt, mTOR, and S6, while overexpressing SALL2 in MCF7-TMR cells had the opposite effect (Fig 5B).

Importantly, we found that both protein and mRNA levels of PTEN, the key negative regulator of Akt/mTOR pathway, were decreased in SALL2-silenced MCF7 cells and upregulated in SALL2-overexpressing MCF7-TMR and ZR-75-30 cells, compared with control cells (Fig 5B and Appendix Fig S4B), suggesting that PTEN

---

**Figure 3. SALL2 transcriptionally activates ESR1.**

A   qRT–PCR analysis of *ESR1* expression in SALL2-silenced, SALL2-overexpressing, and control cells. *GAPDH* was used as an internal control.

B   WB analysis of expression of SALL2 and ERα in the indicated cells. α-Tubulin was used as a loading control.

C   Upper panel: schematic illustration of the predicted binding site for SALL2 in the indicated ESR1 promoter regions (upper panel). Lower left panel: schematic illustration of the wild-type or mutant ESR1 promoter regions cloned into the pGL3 luciferase reporter plasmid; lower right panel: quantification of luciferase activity of the ESR1 promoter reporter was examined in the indicated cells (lower panel). Putative SALL2-binding sites are shown as red filled circles, and the blue filled box shows the mutated site. Red letters in each binding region indicate the putative or mutated SALL2-binding sequences. Vct, empty vector; Wt, wild-type; Mut, mutant.

D   Schematic illustration of the human ESR1 gene promoter (upper panel) and ChIP analysis of enrichment of SALL2 on the ESR1 promoter (lower panel). IgG was used as a negative control. The red squares represent the qRT–PCR region.

E   ChIP assays were performed in the indicated cells using anti-p300 acetyltransferase, anti-RNA POL II (RNAP II), and anti-H3K4me3 antibodies.

F   ERE luciferase assays were performed to assess ERα activation in the indicated cells transfected with ERE-luc (3×ERE) plasmid and then treated with E2 (10 nM) or TAM (1 μM) for 24 h.

G   qRT–PCR analysis of *TFF1* and *PGR* expression in the indicated cell lines treated with or without E2 (10 nM) or TAM (1 μM) for 24 h. *GAPDH* was used as an internal control.

Data information: In (A, left; C–E), data are presented as mean ± SD, and P-values were determined by one-way ANOVA test, n = 3. In (A, middle and right), P-values were determined by two-tailed Student's t-test, n = 3. In (F and G), data are presented as mean ± SD, and P-values were determined by two-way ANOVA test, n = 3. *P < 0.05, **P < 0.01, ***P < 0.001, n.s., no significance. Exact P-values are specified in Appendix Table S10.

Source data are available online for this figure.

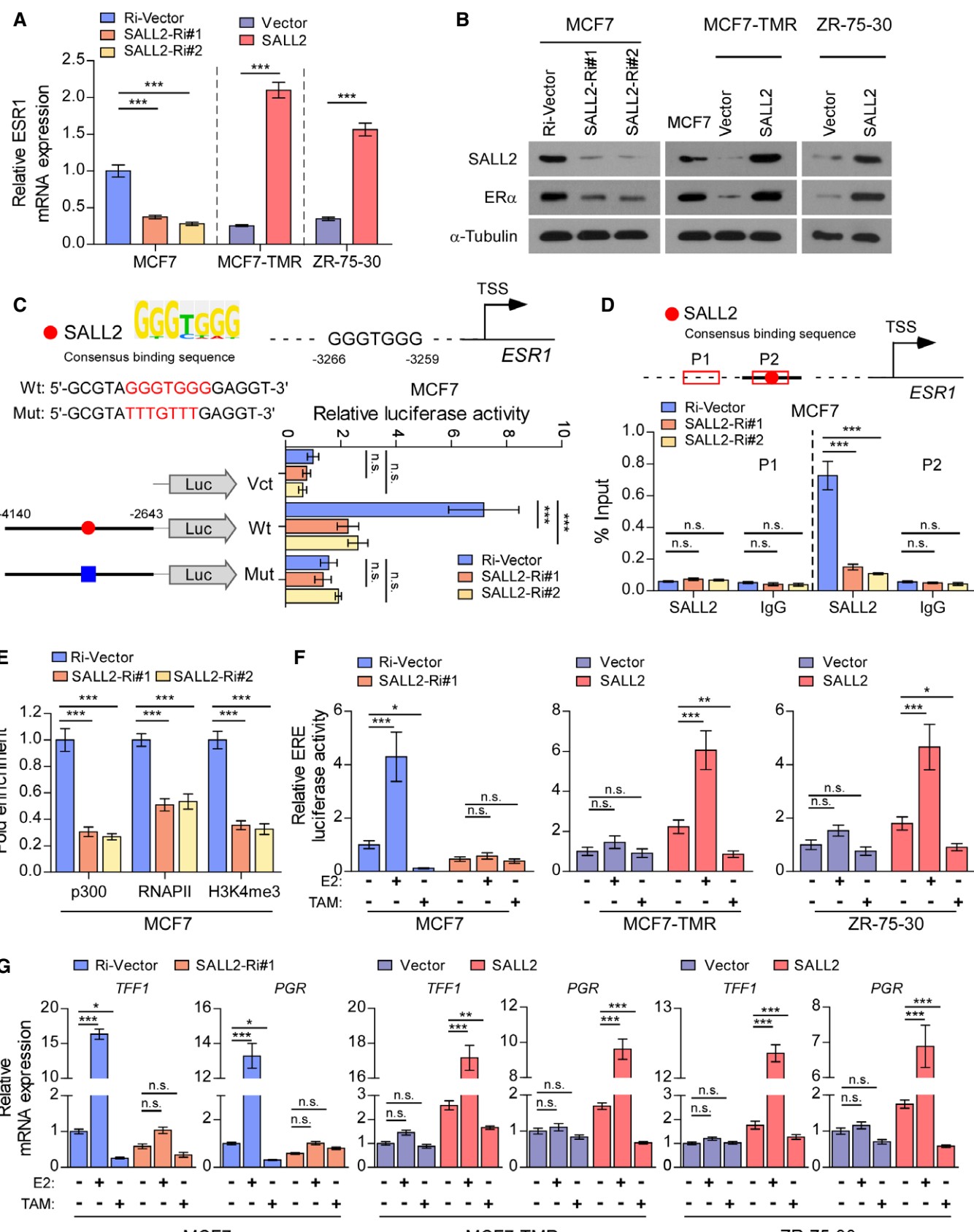

Figure 3.

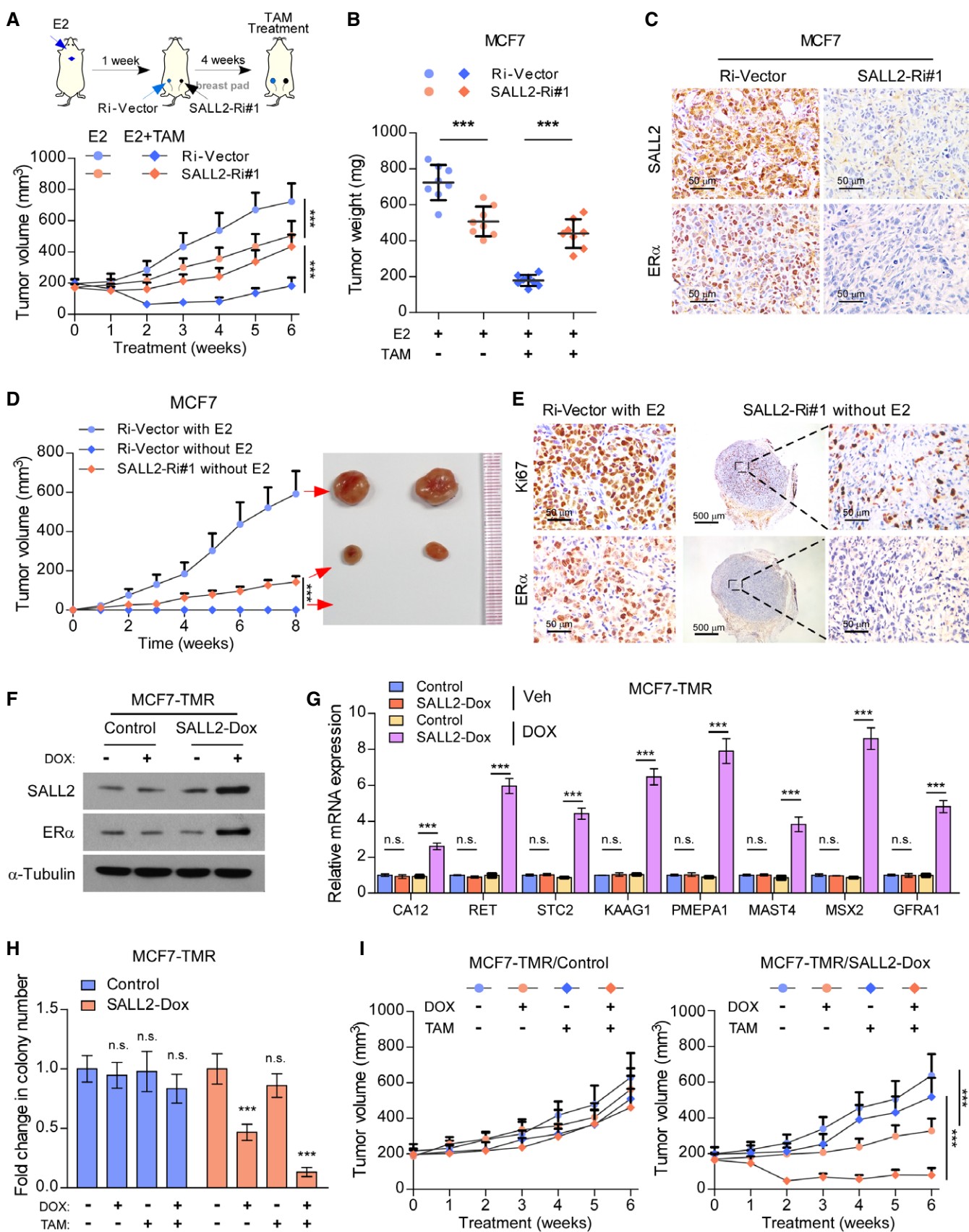

**Figure 4.**

**Figure 4.  Silencing SALL2 promotes tamoxifen resistance via downregulation of ESR1.**

A  Schematic illustration of *in vivo* models of tamoxifen therapy in formed by MCF7 cells (upper panel). Tumor growth curves of the indicated xenograft tumors (*n* = 8/group) (lower panel).

B  Quantification of xenograft tumor weight at the end of the experiment shown in (A) (*n* = 8/group).

C  Representative IHC images of SALL2 and ERα in the E2- and TAM-treated xenografts at the end of the experiment shown in (A). Scale bars: 50 μm.

D  Tumor growth curves of the indicated xenograft tumors (*n* = 8/group) (left panel) and representative tumor images (right panel).

E  Representative IHC images of Ki67 and ERα in the xenografts. Scale bars: 50 μm (left and right panels) and 500 μm (middle panel).

F  WB analysis of the indicated protein expression in the MCF7-TMR/control and MCF7-TMR/SALL2-Dox cells treated with or without doxycycline (Dox); α-tubulin was used as a loading control.

G  qRT−PCR analysis of expression of multiple ERα downstream target genes in the MCF7-TMR/control and MCF7-TMR/SALL2-Dox cells treated with vehicle (Veh) or doxycycline (DOX). *GAPDH* was used as an internal control.

H  Quantification of crystal violet-stained colony formed by the indicated cells treated with or without Dox.

I  Tumor growth curves of the indicated xenograft tumors (*n* = 8/group) upon doxycycline and 5-Aza-dC treatment. 5-Aza-dC treatment was started when the tumors reached approximately 200 mm³.

Data information: In (A, D, H, and I), data are presented as mean ± SD, and *P*-values were determined by two-way ANOVA test, *n* = 3 in (H). In (B and G), data are presented as mean ± SD, and *P*-values were determined by 1-way ANOVA, *n* = 3 in (G). ***P* < 0.001, n.s., no significance. Exact *P*-values are specified in Appendix Table S10.

Source data are available online for this figure.

might be a potential downstream target of SALL2 in breast cancer cells. Luciferase reporter assays showed that silencing SALL2 repressed, while overexpression of SALL2 increased, the reporter activity of PTEN promoter containing the first putative SALL2-binding site (Appendix Fig S4C). Furthermore, ChIP assays indicated that endogenous SALL2 bound to the promoter region of PTEN at the first binding site, and the enrichment of p300, RNA POL II, and H3K4me3 on this promoter region was significantly decreased in SALL2-silenced cells (Appendix Fig S4D and E). These results demonstrate that SALL2 increases PTEN transcription by binding to its promoter. Consistently, MCF7-TMR/SALL2-Dox cells treated with doxycycline induced PTEN expression but decreased phosphorylated Akt (Appendix Fig S4F). Moreover, we found that PTEN inhibition with the inhibitor SF1670 increased the level of phosphorylated Akt in SALL2-overexpressing MCF7-TMR cells and abrogated SALL2 overexpression effect on cell viability and apoptosis in MCF7-TMR cells (Fig 5C and D). Taken together, our results indicate that SALL2 silencing activates Akt/mTOR pathway via downregulation of PTEN.

Furthermore, ipatasertib (GDC-0068), an Akt inhibitor currently used in clinical breast cancer therapy (Lin *et al*, 2013; Kim *et al*, 2017), was used to examine the crucial effect of Akt/mTOR signaling pathway on SALL2 downregulation-induced tumorigenicity of ER$^+$ breast cancer cells. As shown in Fig 5E–G, administration of ipatasertib significantly inhibited the tumorigenic capability of estrogen-

treated MCF7/SALL2-Ri#1 cells. More importantly, ipatasertib treatment completely abrogated the tumor formation capacity of MCF7/SALL2-Ri#1 cells without estrogen pellet implantation (Fig 5E and F). Thus, these results indicate that PI3K/Akt pathway is required for SALL2 knockdown-induced tumor growth independent of estrogen.

**The SALL2 promoter is frequently methylated in tamoxifen-resistant breast cancer**

It has been reported that promoter methylation-mediated SALL2 downregulation occurs in multiple cancers (Sung *et al*, 2013; Luo *et al*, 2017; Imai *et al*, 2019). We also found that treatment with 5-Aza-2′-deoxycytidine (5-Aza-dC), a DNA methyltransferase inhibitor, robustly increased the transcript level of SALL2 in MCF7-TMR cells, but had no effect on MCF7 cells or tamoxifen-treated MCF7 cells (Fig 6A and B). Importantly, bisulfite genomic sequencing PCR (BSP) analysis revealed that DNA methylation density on SALL2 promoter region in MCF7-TMR cells was significantly higher than that in MCF7 parental cells, suggesting that SALL2 methylation in breast cancer cells was increased during tamoxifen therapy (Fig 6C and D). ChIP assays revealed that enrichment of DNMT1 and DNMT3B, but not DNMT3A, and recruitment of H3K27me3 and H3K9me2 were increased in MCF7-TMR cells (Fig 6E).

In breast cancer cell lines and tissues, SALL2 hypermethylation status was significantly correlated with low SALL2

**Figure 5.  Silencing SALL2 promotes cell survival via activating Akt/mTOR signaling.**

A  GSEA profiling of correlation between low SALL2 expression and PI3K/Akt/mTOR signaling signature using the TCGA (BRCA) dataset. The blue bars showed the NES (normalized enrichment score). The black line is the negative of the q value (FDR, false discovery rate).

B  WB analysis of the indicated protein expression in the cells treated with E2 (10 nM) or TAM (1 μM) for 24 h. α-Tubulin was used as a loading control.

C  WB analysis of indicated protein expression in MCF7-TMR cells treated with vehicle or PTEN inhibitor SF1670. α-Tubulin was used as a loading control.

D  Viability (left panel) or Annexin V–FITC/PI staining (right panel) of the cells treated with vehicle or PTEN inhibitor SF1670.

E  Tumor growth curves of the MCF7/SALL2-Ri#1 xenograft tumors (*n* = 8/group) upon the indicated treatment. Ipatasertib treatment was started when the tumors grew for 4 weeks.

F  Quantification of xenograft tumor weight at the end of the experiment shown in (E) (*n* = 8/group).

G  Representative IHC images and quantification of Ki67 and TUNEL staining of apoptotic cells in the vehicle (Veh)- or ipatasertib-treated MCF7/SALL2-Ri#1/xenografts (*n* = 8/group). Scale bars: 50 μm.

Data information: In (D and F), data are presented as mean ± SD, and *P*-values were determined by one-way ANOVA test, *n* = 3 in (D). In (E), data are presented as mean ± SD, and *P*-values were determined by two-way ANOVA test. In (G), data are presented as mean ± SD, and *P*-values were determined by two-tailed Student's *t*-test. ***P* < 0.01, ****P* < 0.001. Exact *P*-values are specified in Appendix Table S10.

Source data are available online for this figure.

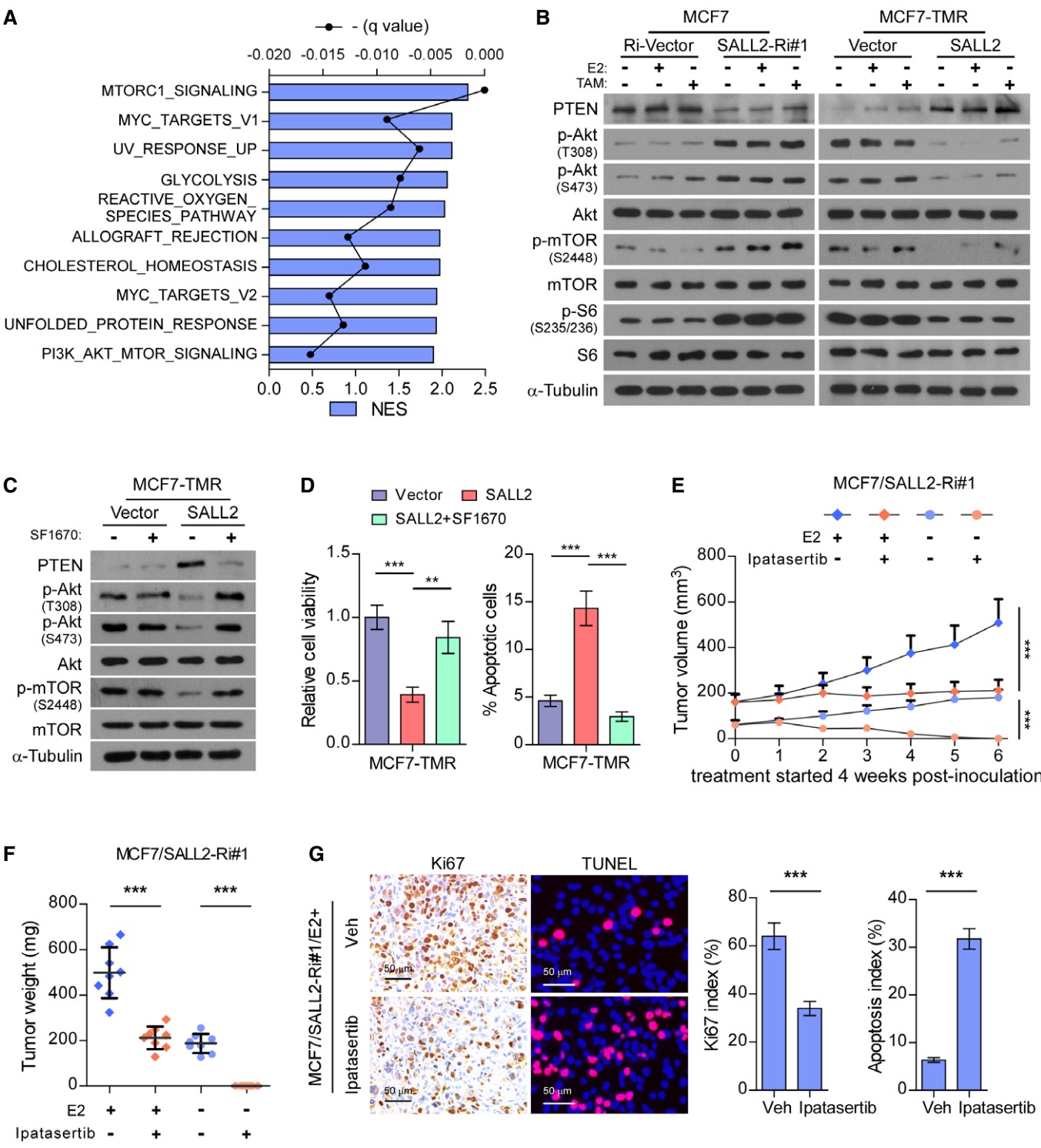

Figure 5.

expression ($P < 0.001$; $n = 90$; Fig 6F and G). Importantly, SALL2 hypermethylation status was positively associated with shorter DFS and OS in breast cancer patients (Fig 6H). Altogether, these results indicate that promoter hypermethylation mediates SALL2 downregulation in tamoxifen-resistant breast cancer.

**SALL2 increases sensitivity to tamoxifen in ESR1-hypomethylated ER⁻ breast cancer**

The promoter region of the ESR1 gene was found to be frequently hypermethylated in ER-negative breast cancer tissues, which is one of the mechanisms mediating ER downregulation

(Lapidus *et al*, 1998; Yang *et al*, 2001). Interestingly, we found that SALL2 mRNA expression was significantly associated with ESR1 level in ER-hypomethylated ($P = 0.005$, $r = 0.420$; $n = 44$) but not in ER-hypermethylated ($P = 0.201$, $r = 0.130$; $n = 99$) ER-negative breast cancer tissues (Fig EV4A). Importantly, overexpression of SALL2 significantly increased ER mRNA in ER-

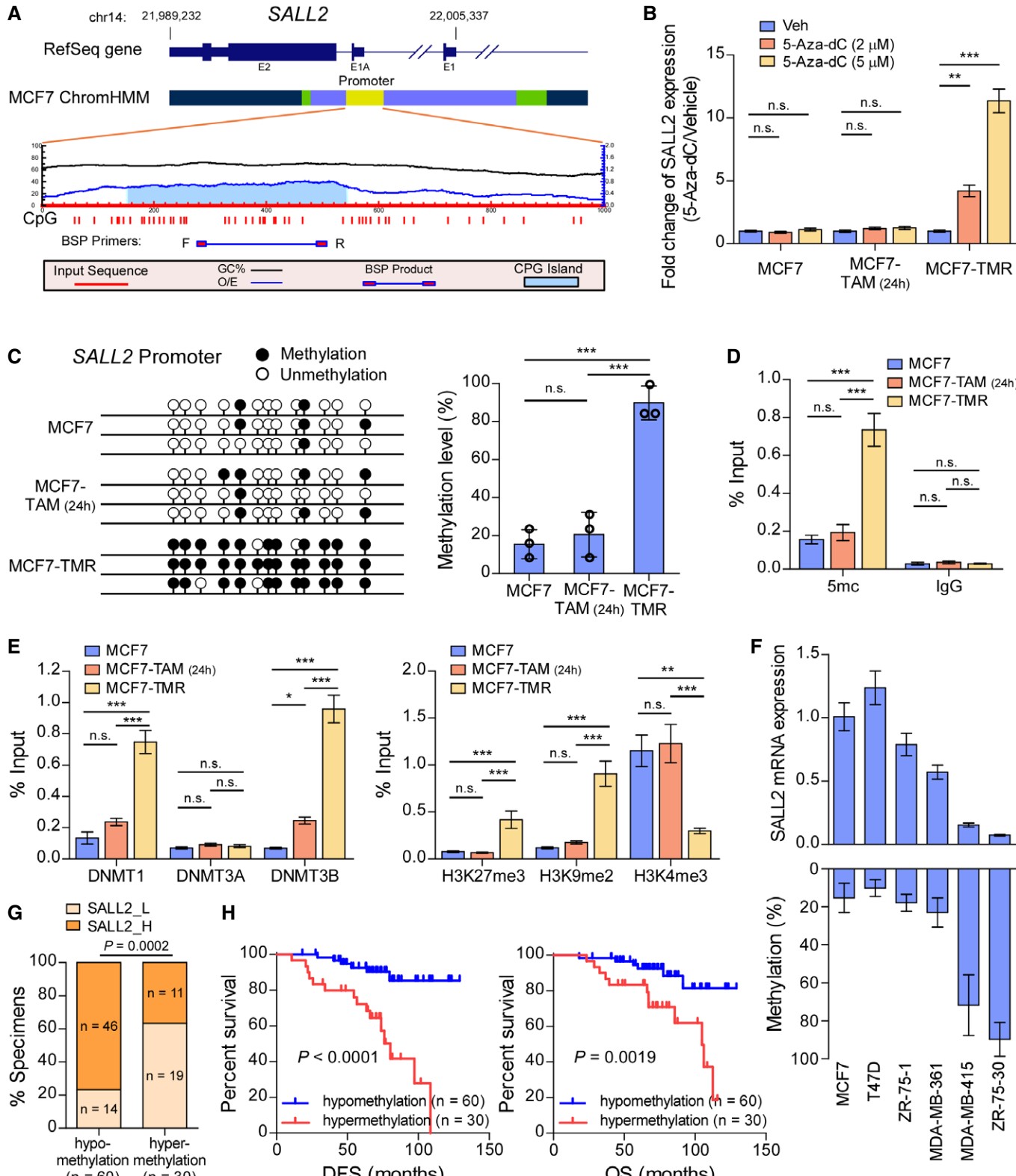

Figure 6.

◄

**Figure 6. SALL2 promoter is methylated in tamoxifen-resistant breast cancer.**

A  Color coding of the ChromHMM regions in MCF7 cells obtained from GEO (GSE69118): yellow, promoter; green, enhancer; light blue, transcribed; and blue, un-transcribed. BSP primers were designed to amplify the sequence in the predicted CpG island region.
B  qRT–PCR analysis of *SALL2* expression in the indicated cells treated with vehicle or 5-Aza-dC. *GAPDH* was used as an internal control.
C  BSP analysis (left panel) and quantification (right panel) of the methylation status of *SALL2* gene in the indicated cells.
D, E  ChIP analyses of enrichment of 5mc (D) and DNMTs and histone modifications (E) on the SALL2 promoter.
F  qRT–PCR analysis of *SALL2* expression (upper panel) and BSP analysis of SALL2 methylation status (lower panel) in 6 ER$^+$ breast cancer cell lines. *GAPDH* was used as an internal control.
G  Bar graph showing negative correlation of SALL2 DNA methylation level with SALL2 expression analyzed by IHC staining.
H  Kaplan–Meier analysis of DFS (left panel) or OS (right panel) curves in tamoxifen-treated patients with SALL2-hypomethylated and SALL2-hypermethylated ER$^+$ breast cancer.

Data information: In (B–E), data are presented as mean $\pm$ SD, and P-values were determined by one-way ANOVA test, $n = 3$. In (F), data are presented as mean $\pm$ SD, $n = 3$. In (G), $n = 90$, P-values were determined by $\chi^2$ test. In (H), $n = 90$, log-rank test. *$P < 0.05$, **$P < 0.01$, ***$P < 0.001$, n.s., no significance. Exact P-values are specified in Appendix Table S10.

hypomethylated ER-negative breast cancer cells (BT-20 cells), but had no effect on ER expression in ER-hypermethylated ER-negative breast cancer cells (BT-549 and MDA-MB-231 cells; Fig EV4B and C). Moreover, overexpression of SALL2 in BT-20 cells, but not in BT-549 and MDA-MB-231 cells, resulted in significantly increased tamoxifen response (Fig EV4D). These results further support the notion that SALL2 activates ER expression and enhances tamoxifen response in breast cancer.

### DNMT inhibitor treatment resensitizes SALL2-hypermethylated tamoxifen-resistant breast cancer cells to tamoxifen therapy *in vivo*

We next examined the effect of DNMT inhibitor on tamoxifen sensitivity in SALL2-hypermethylated breast cancer cells in an *in vivo* mouse model. As shown in Fig 7A and B, 5-Aza-dC treatment significantly increased the anti-tumor effect of tamoxifen on tumors formed by SALL2-hypermethylated MCF7-TMR cells. Consistently, 5-Aza-dC-treated MCF7-TMR xenografts displayed reduced Ki67 levels, higher apoptotic rates, and increased expression of SALL2, ERα, and PTEN compared with control tumors (Fig 7C–E). Therefore, these results indicated that DNMT inhibitor treatment resensitized SALL2-hypermethylated tamoxifen-resistant breast cancer to tamoxifen therapy *in vivo*. Furthermore, as shown in Fig EV5A and B, 5-Aza-dC treatment did not induce ESR1 expression in SALL2-silenced MCF7/TMR and MCF7 cells and failed to abrogate the tumorigenic capability of SALL2-silenced MCF7/TMR and MCF7 cells with or without tamoxifen treatment *in vivo* (Figs 7F–H and EV5C–E), which confirmed that SALL2 is indispensable for 5-Aza-dC-induced tamoxifen resensitization.

## Discussion

Tamoxifen, which targets ER, is still the first-line therapy and most widely used treatment for ER-positive breast cancer. However, resistance to tamoxifen therapy associated with disease progression is a significant clinical challenge (Osborne & Schiff, 2011; Rondon-Lagos *et al*, 2016). Loss of ER expression is the dominant mechanism of *de novo* resistance to tamoxifen (Osborne & Schiff, 2011; Rondon-Lagos *et al*, 2016). In the present study, we further found that the transcription factor SALL2 was downregulated during tamoxifen therapy via profiling of 9 paired breast cancer tissues (pre-tamoxifen primary tumors and matched relapsed tamoxifen-resistant tissues

from the same individuals). By performing a series of *in vitro* and *in vivo* experiments, we further demonstrated that SALL2 was a key upstream regulator of ER expression. Moreover, SALL2 hypermethylation status significantly correlated with a shorter disease-free survival time in tamoxifen-resistant breast cancer patients. Restoration of SALL2 with DNMT inhibitor increased sensitivity to tamoxifen therapy in SALL2-hypermethylated ER-positive and SALL2-hypermethylated/ESR1-hypomethylated ER-negative breast cancer cells. Therefore, our results shed light on the role of SALL2 in ER regulation and identify a potential clinical biosignature that could be used for subgrouping breast cancer patients and identify those who might benefit from tamoxifen/DNMT inhibitor co-therapy.

SALL2, a member of Spalt-like transcription factor family, plays important roles in various biological processes, such as neurogenesis, neuronal differentiation, kidney development, and eye morphogenesis (Bohm *et al*, 2008; Chatterjee *et al*, 2013; Kelberman *et al*, 2014). SALL2 also contributes to cellular apoptosis, growth arrest, and quiescence maintenance by regulation of multiple downstream genes, such as p21Cip1/Waf1, p16Ink4a, c-MYC, BAX, and Noxa (Dawei Li *et al*, 2004; Gu *et al*, 2011; Sung *et al*, 2012; Escobar *et al*, 2015; Wu *et al*, 2015). Recently, SALL2 was found downregulated in various types of cancer, including leukemia, ovarian, lung, and radioresistant esophageal cancers (Sung *et al*, 2013; Liu *et al*, 2014; Luo *et al*, 2017), suggesting that SALL2 may act as a tumor suppressor. However, Suva *et al* (2014) found that SALL2 may also function as an oncogene, as it contributed to converting differentiated glioblastoma cells into cancer stem-like cells, thus favoring glioblastoma propagation. These apparently contradictory results suggest that the functional roles of SALL2 may differ according to the cell type and cellular context. In the current study, we demonstrated that SALL2 could simultaneously upregulate ERα and PTEN through direct binding to their promoters, suggesting that SALL2 may have dual functions in breast cancer cells. Therefore, the precise molecular mechanism by which SALL2 mediates transcriptional upregulation of ERα and PTEN in breast cancer cells needs further investigation.

In the current study, we identified that SALL2 acted as a key upstream regulator of ER expression, and expression of SALL2 was significantly downregulated during tamoxifen therapy. Importantly, doxycycline-inducible mouse model revealed that restoration of SALL2 sensitized tamoxifen-resistant breast tumor to tamoxifen therapy. Therefore, SALL2 could be a potential target in tamoxifen-resistant breast cancer in the future, as it is currently not possible to clinically upregulate SALL2 via transferring SALL2 cDNA into breast

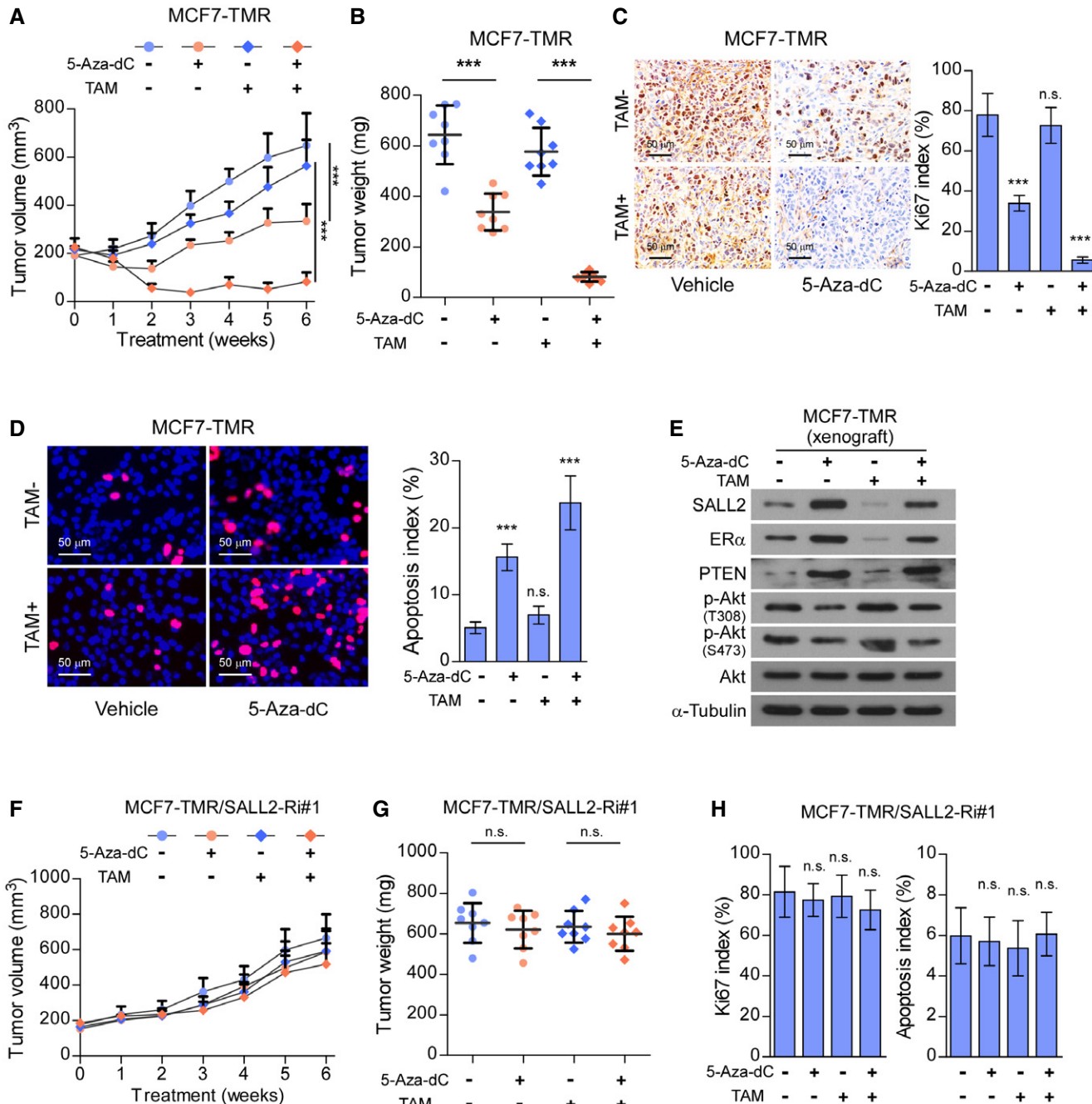

**Figure 7. Upregulation of SALL2 restores the sensitivity of resistant breast cancer cells to tamoxifen *in vivo*.**

A   Tumor growth curves of MCF7-TMR/xenografts (*n* = 8/group) upon the indicated treatment.

B   Quantification of xenograft tumor weight at the end of the experiment shown in (A) (*n* = 8/group).

C   Representative images (left panel) and quantification (right panel) of IHC analysis of Ki67 signals in the MCF7-TMR/xenografts upon the indicated treatment
    (*n* = 8/group). Scale bars: 50 μm.

D   Representative images (left panel) and quantification (right panel) of TUNEL analysis of apoptotic cells in the MCF7-TMR/xenografts upon the indicated treatment
    (*n* = 8/group). Scale bars: 50 μm.

E   WB analysis of the indicated protein expression in the MCF7-TMR/xenografts upon the indicated treatment.

F   Tumor growth curves of xenografts (*n* = 8/group) formed by MCF7-TMR/SALL2-Ri cells upon the indicated treatment.

G   Quantification of xenograft tumor weight at the end of the experiment shown in (F) (*n* = 8/group).

H   IHC analysis of Ki67 signals (left panel) and TUNEL analysis of apoptotic cells (right panel) in the MCF7-TMR/xenografts upon the indicated treatment (*n* = 8/group).

Data information: In (A and F), data are presented as mean ± SD, and *P*-values were determined by two-way ANOVA test. In (B–D, G, and H), data are presented as
mean ± SD, and *P*-values were determined by one-way ANOVA test. ***P* < 0.001, n.s., no significance. Exact *P*-values are specified in Appendix Table S10.
Source data are available online for this figure.

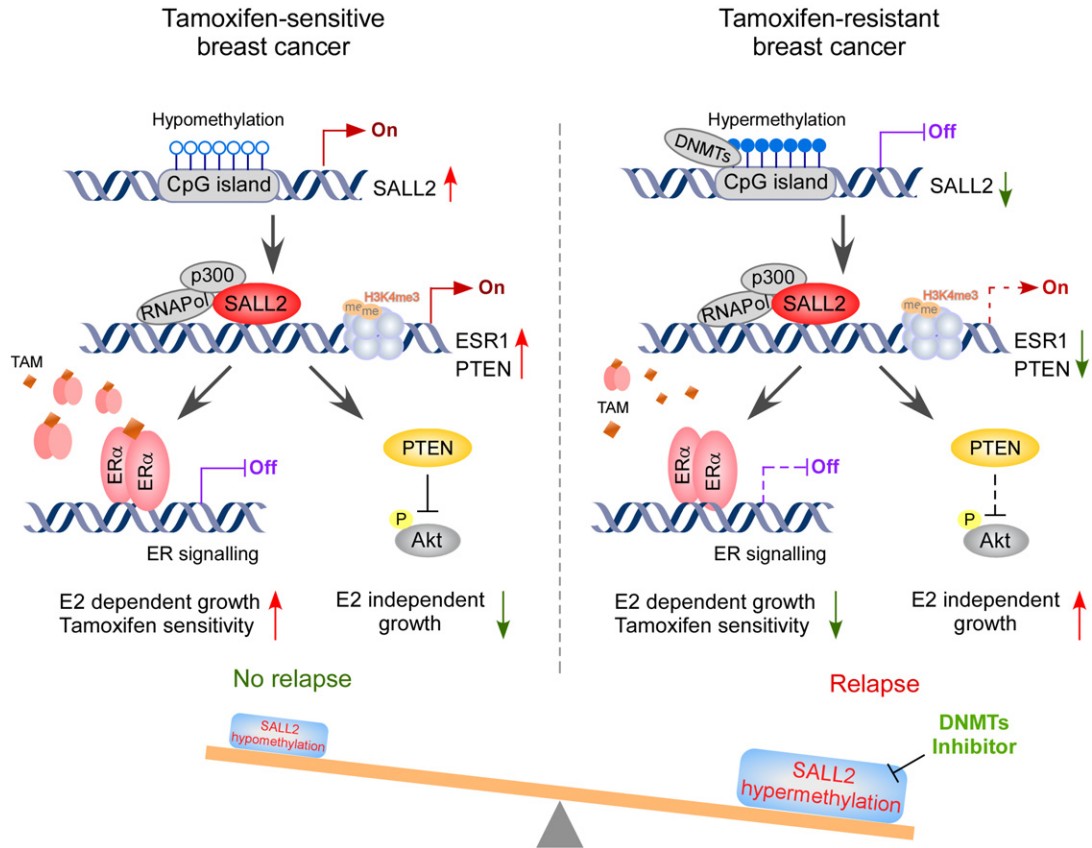

**Figure 8.  Model of promoter methylation-mediated SALL2 reduction confers tamoxifen resistance in ER+ breast cancer.**

In SALL2-hypomethylated ER+ breast cancer, SALL2 activates ER signaling via transcriptional upregulation of ESR1 and inhibits Akt/mTOR signaling via transcriptional upregulation of PTEN (left), leading to an estrogen-dependent growth and tamoxifen sensitivity. In SALL2-hypermethylated ER+ breast cancer, promoter methylation-mediated SALL2 reduction represses ERα and PTEN expression and activates Akt/mTOR signaling, resulting in estrogen-independent tumor growth and tamoxifen-resistant phenotype. DNMT inhibitor-induced SALL2 restoration resensitizes tamoxifen-resistant breast cancer to endocrine therapy (right).

tumors. We further investigated the mechanism by which SALL2 was downregulated in breast cancer. Our results showed that promoter hypermethylation-mediated SALL2 downregulation conferred resistance to ER-positive breast cancer cells. Treatment with 5-Aza-dC, a DNMT inhibitor, increased sensitivity of SALL2-hypermethylated breast cancer to tamoxifen therapy. Hence, our results suggest that treatment with a DNMT inhibitor might overcome tamoxifen resistance in breast cancer. It has been reported that only 30% of breast cancers are linked to epigenetic modification related to DNA methylation (Fan et al, 2006; Novak et al, 2008; Shann et al, 2008). Therefore, co-therapy including both a DNMT inhibitor and tamoxifen might be an appropriate therapy only for a subset of patients with breast cancer. Meanwhile, more AIs- or SERDS-treated breast cancer samples should be collected to further examine the correlation between SALL2 methylation status and SALL2 low expression. Studying the clinical outcome from an alternative therapy (e.g., aromatase inhibitor or SERDS) in these samples might help finding a potential clinical biomarker.

Hyperactivation of PI3K/Akt/mTOR pathway has been considered as essential element leading to endocrine therapy resistance in breast cancer (Miller et al, 2011). Cumulative data from clinical studies support the hypothesis that Akt hyperphosphorylation in

breast cancer predicts a worse outcome among endocrine-treated patients (Perez-Tenorio & Stal, 2002; Osborne & Schiff, 2011). Currently, the U.S. FDA has approved a combination of exemestane (AIs) and everolimus (TORC1 inhibitors) for the treatment of advanced ER+ breast cancers. However, other studies suggest that while inhibition of the PI3K/Akt pathway results in reduced cell proliferation and lower survival, compensatory feedback events within this pathway confer resistance to single inhibitors (deGraffenried et al, 2004; Lui et al, 2016). In such a model, Akt inhibition activates positive feedback loops driven by MYC and resulting in increased gene expression of ERBB2, ERBB3, ERK5, and IGF1 (Ribas et al, 2015). These observations have triggered investigations into the mechanisms regulating Akt signaling in endocrine-resistant breast cancer. In the present study, we found that PTEN was transcriptionally upregulated by SALL2. Downregulation of SALL2 led to PTEN reduction and Akt activation, and therefore contributed to the survival of breast cancer cells treated with tamoxifen. Remarkably, restoration of SALL2 maintained the expression of PTEN, which provided a novel strategy to inhibit Akt signaling, resulting in suppression of the breast cancer cell survival. Thus, our results confirm that activation of the Akt/mTOR pathway contributes to tamoxifen resistance in breast cancer, but also reveal a novel mechanism

for regulation of the Akt/mTOR signaling pathway involving the SALL2–PTEN axis.

In summary, our study identified that SALL2, as a novel upstream regulator of ERα, modulated tamoxifen resistance in ER$^+$ breast cancer (Fig 8). Low SALL2 expression or SALL2 hypermethylation status predicted higher risk of metastatic relapse and poor prognosis. Mechanistically, hypermethylation-mediated SALL2 reduction induced anti-estrogen resistance by downregulating ERα and promoted hormone-independent growth by regulation of PTEN/Akt/mTOR signaling cascade in human breast cancer cells. Our findings shed light on the role of SALL2 in the transcriptional regulation of ER, and present a potential clinical biosignature that could be used for subgrouping breast cancer patients and identify those who might benefit from co-therapy with tamoxifen and DNMT inhibitor.

# Materials and Methods

## Cell culture

Human breast cancer cell lines MCF7, T47D, and MDA-MB-231 were cultured in DMEM with 10% fetal bovine serum (FBS); ZR-75-1, ZR-75-30, and BT-549 cells were cultured in RPMI 1640 with 10% FBS; MDA-MB-361 and MDA-MB-415 cells were cultured in Leibovitz's L-15 medium with 10% FBS; and BT-20 cells were cultured in EMEM with 10% FBS. The above cell lines were purchased from American Type Culture Collection (ATCC, Manassas, VA, USA). All cell lines were authenticated using short tandem repeat (STR) profiling. Cells were maintained at 37°C in a 5% $CO_2$ incubator. For β-estradiol (E2, #E2758; Sigma-Aldrich) and 4-hydroxytamoxifen (4-OH-TAM, abbreviated as TAM, #H6278; Sigma-Aldrich) treatment, cells were cultured in phenol red-free DMEM supplemented with 10% charcoal/dextran-treated FBS. Tamoxifen-resistant MCF7 cells (MCF7-TMR) cells were established in the E2-deficient media but supplemented with a 10-fold increase in the TAM concentration ($10^{-9}$–$10^{-6}$ M) every 4 weeks. Authentication by STR profiling was performed to ensure the MCF7-TMR cell line was derived from MCF7 cell line, which was maintained with TAM ($10^{-6}$ M) (Herman & Katzenellenbogen, 1996). Cells were routinely tested for mycoplasma contamination using the LookOut Mycoplasma PCR Detection Kit (#MP0035; Sigma-Aldrich).

## Constructs and transfection

Human SALL2 and ESR1 cDNAs were PCR-amplified and cloned into the pMSCV-puro-retro vector (#634401; TaKaRa). The shRNAs in vector pLKO.1-puro were purchased from Transheep Bio. The luciferase cDNA was PCR-amplified and cloned into the pMSCV-neo-retro vector. The promoter sequences were cloned into the pGL3 luciferase reporter plasmid (#E1751; Promega, Madison, WI, USA). The promoter reporter constructs with mutant SALL2-binding motif were synthesized using a site-specific mutagenesis kit (#210518; Stratagene, La Jolla, CA, USA). The luciferase reporters 3 × ERE (estrogen-responsive element) TATA luc was purchased from Addgene (plasmid #11354). Cells ($2 \times 10^5$) were seeded and infected by retrovirus generated by pMSCV-puro-cDNAs or with lentivirus produced by pLKO.1-puro-shRNAs for 3 days. All cells were infected with a pMSCV-neo-luciferase retrovirus and selected with 0.5 μg/ml puromycin and 250 μg/ml G418 for 7 days to

establish stable cell lines. Transfection of luciferase reporter plasmids was performed using the Lipofectamine 3000 reagent (#L3000015; Invitrogen, Carlsbad, CA, USA) according to the manufacturer's instructions.

## Human tissue specimens

A total of 238 archived formalin-fixed, paraffin-embedded breast cancer specimens, which were histopathologically diagnosed at the Sun Yat-sen University Cancer Center from 2004 to 2012, were collected for this study. Nine paired tissue samples from patients with relapsed breast cancer were collected pre- and post-tamoxifen treatment for RNA-seq analysis. Informed consent was obtained from all subjects, and the experiments conformed to the principles set out in the WMA Declaration of Helsinki and the Department of Health and Human Services Belmont Report.

## RNA-sequencing (RNA-seq) analysis

RNA-seq analysis was performed using nine paired breast cancer samples, comprising the primary tissue samples at diagnosis and the matched relapsed tamoxifen-resistant tissue samples. Appropriate information is summarized in Appendix Table S1. Gross dissection was performed to slice up a core tissue that contained tumor-rich areas. Total RNA was extracted from cultured cells and tissues using the TRIzol reagent (#15596018; Invitrogen; Life Technologies, USA) according to the manufacturer's instructions. The RNA purity, concentration, and integrity were evaluated using a NanoPhotometer® spectrophotometer (IMPLEN, Westlake Village, CA, USA), Qubit® RNA Assay Kit in Qubit® 3.0 Fluorometer (Life Technologies, Carlsbad, CA, USA), and RNA Nano 6000 Assay Kit of the Bioanalyzer 2100 system (Agilent Technologies, Santa Clara, CA, USA), respectively. The mRNAs isolated from the total RNA were fragmented into short fragments using fragmentation buffer (#AM8740; ThermoFisher). Double-stranded cDNA was synthesized using these short fragments as templates. The cDNA was end-repaired, ligated to Illumina adapters, size-selected on an agarose gel (approximately 250 bp), and PCR-amplified. Following the recommendations of the manufacturer, the cDNA library for sequencing was generated using a VAHTSTM mRNA-seq v2 Library Prep Kit (Vazyme Biotech Co., Ltd, Nanjing, China). Libraries were sequenced on an Illumina HiSeq X Ten platform (Illumina Inc., San Diego, CA, USA) using the (2 × 150 bp) paired-end module. After initial quality control, the clean reads were mapped to the human genome (version hg38) using TopHat2 with a reference annotation from the Ensembl database (Kim et al, 2013). The gene expression profile of each sample was analyzed quantitatively using the Cuffdiff suite from Cufflinks (v2.2.1) (Trapnell et al, 2012), a program for the comparative assemblies of transcripts and the estimation of their abundance in a transcriptome sequencing experiment, which uses the measurement unit FPKM (fragments per kilobase of transcript per million mapped reads).

## Western blotting (WB) analysis

WB analysis was performed as described previously (Li et al, 2008). The protein concentration was determined using a bicinchoninic acid (BCA) assay (#23225; Pierce, Rockford, IL, USA) according to

the manufacturer's instructions. The following antibodies were used in this study: anti-SALL2 Rabbit antibody (1:500 dilution; #A303-208A; Bethyl), anti-NKX3-1 Mouse antibody (1:1,000 dilution; #NB100-1828; Novus), anti-ERα Rabbit antibody (1:1,000 dilution; #HPA000449; Sigma-Aldrich), anti-PTEN rabbit antibody (1:1,000 dilution; #9188; Cell Signaling Technology), anti-Akt mouse antibody (1:2,000 dilution; #2920; Cell Signaling Technology), anti-phospho-Akt (Ser473) rabbit antibody (1:2,000 dilution; #4060; Cell Signaling Technology), anti-phospho-Akt (Thr308) rabbit antibody (1:1,000 dilution; #9275; Cell Signaling Technology), anti-mTOR Rabbit antibody (1:1,000 dilution; #2983; Cell Signaling Technology), anti-Phospho-mTOR (Ser2448) Rabbit antibody (1:1,000 dilution; #5536; Cell Signaling Technology), anti-S6 Ribosomal Protein Rabbit antibody (1:1,000 dilution; #2317; Cell Signaling Technology), anti-Phospho-S6 Ribosomal Protein (Ser235/236) Rabbit antibody (1:2,000 dilution; #2211; Cell Signaling Technology), anti-α-tubulin mouse monoclonal antibody (1:4,000 dilution; #NB120-11325; Novus), and the secondary antibodies goat anti-rabbit immunoglobulin G (1:2,000 dilution; #ab7090; Abcam) and goat anti-mouse immunoglobulin G (1:2,000 dilution; #ab97040; Abcam).

### Immunohistochemistry analysis

Immunohistochemistry analysis was performed on the 238 paraffin-embedded breast cancer tissues using Histostain-Plus Kits (#859043; ThermoFisher) following the manufacturer's protocols. Anti-SALL2 rabbit polyclonal antibody (1:100 dilution; #HPA004162; Sigma-Aldrich), anti-ERα rabbit polyclonal antibody (1:100 dilution; #HPA000449; Sigma-Aldrich), and anti-Ki67 mouse monoclonal antibody (1:400 dilution; #9449; Abcam) were used for IHC staining. IHC staining of SALL2 was scored separately by two independent pathologists. Tumor cell proportions were scored as follows: 0, no positive tumor cells; 1, < 10% positive tumor cells; 2, 10–35% positive tumor cells; 3, 35–75% positive tumor cells; and 4, > 75% positive tumor cells. Staining intensity was graded according to the following standard: 0, no staining; 1, weak staining (light yellow); 2, moderate staining (dark yellow/light brown); and 3, strong staining (dark brown). The staining index (SI) was calculated as the product of the staining intensity score and the proportion of positive tumor cells. Using this method of assessment, we evaluated SALL2 protein expression in breast cancer tissues by determining the SI, with possible scores of 0, 1, 2, 3, 4, 6, 8, 9, and 12. An optimal threshold of SI ≥ 6 was then determined to define tumors with high expression using the Cutoff Finder program (http://molpath.charite.de/cutoff/).

### Quantitative real-time reverse transcription PCR (qRT–PCR)

A total of 1 μg of RNA from each sample was reverse-transcribed to cDNA using M-MLV Reverse Transcriptase (#M1701; Promega, Madison, WI, US). qRT–PCR analysis was performed on a CFX96 Real-Time System C1000 Cycler (Bio-Rad Laboratories, Singapore) using TB Green Fast qPCR Mix (#RR430A; TaKaRa). Expression data were normalized to the housekeeping gene GAPDH and calculated as $2^{-[(\text{Ct of gene}) - (\text{Ct of GAPDH})]}$, in which Ct represents the cycle threshold for each transcript. Primers used in the PCRs are listed in Appendix Table S7.

### Cell viability assay

The indicated cells ($2.5 \times 10^3$) were plated per well in 96-well plates. MTT assay solution (#11465007001, Cell Proliferation Kit I; Roche) was added to each well, and the cells were incubated for 2 h. The medium was then aspirated, and cells were resuspended in 200 μl of DMSO (#D2650; Sigma-Aldrich). Absorbance at 560 nm was measured; background read at 670 nm was subtracted. The survival percentage of drug-treated cells versus time-matched vehicle-treated cells was calculated.

### Flow cytometry (FACS) assay

The indicated cells were harvested, fixed in 75% ethanol, and stored at 4°C overnight for later cell cycle analysis using flow cytometry. The fixed cells were centrifuged at 1,000 g for 5 min and washed twice with cold 1× PBS. RNase A (20 μg/ml final concentration) and propidium iodide (50 μg/ml final concentration) were added to the cells and incubated for 30 min at 37°C in the dark. Then, $2 \times 10^5$ cells were analyzed using a FACSCalibur instrument (BD Biosciences) equipped with CellQuest 3.3 software. ModFit LT 3.1 trial cell cycle analysis software was used to determine the percentage of cells in the different phases of the cell cycle. For cell apoptosis assay, Annexin V–FITC/PI Apoptosis Detection Kit (#KGA108-1; KeyGEN BioTECH) was used according to the manufacturer's instructions. Briefly, $2 \times 10^6$ cells were collected and resuspended in binding buffer, stained with fluorochrome-conjugated Annexin V and propidium iodide, and analyzed by flow cytometry (Beckman Coulter, USA).

### Colony formation assay

Colony formation assay was performed to determine cell viability. The indicated cells ($1 \times 10^3$) were plated in 6-well plates. Two weeks later, the surviving colonies were fixed, stained with crystal violet stain, and counted.

### Terminal deoxynucleotidyl transferase nick-end labeling (TUNEL) assay

The TUNEL staining was performed using DeadEnd™ Fluorometric TUNEL System (#G3250; Promega) according to the manufacturer's protocol. Briefly, the sections were deparaffinized, rehydrated, heated in a microwave oven for 1 min, and then immersed in Tris–HCl, 0.1 M pH 7.5, containing 3% BSA and 20% normal bovine serum for 30 min at room temperature. TUNEL reaction mixture was then added to the sections for 60 min at 37°C. In each case, 500–1,000 cells were counted and the mean apoptotic index was calculated.

### Promoter methylation analysis

Genomic DNA extracted from breast cancer cell lines or tissues was bisulfite-modified using an EpiTect Bisulfite Kit (#59104; Qiagen) according to the manufacturer's instructions, and then, bisulfite genomic sequencing polymerase chain reactions (BSP) were conducted using the primers listed in Appendix Table S8. The PCR products were cloned to the pGEM-T Easy Vector System and

sequenced using the M13F universal primer. Three clones for each cell line or each tissue were analyzed, and the average methylation frequency was counted.

## Luciferase activity assay

The indicated cells ($3 \times 10^3$) were cultured in triplicate 48-well plates for 24 h. Then, 100 ng of luciferase reporter plasmids or the control luciferase plasmid, plus 1 ng pRL-TK Renilla plasmid (#E2231; Promega), was transfected into cells using the Lipofectamine 3000 reagent (#L3000015; Invitrogen) according to the manufacturer's recommendations. Luciferase and Renilla signals were measured 24 h after transfection using the Dual Luciferase Reporter Assay Kit (#E1960; Promega).

## Chromatin IP (ChIP) assay

The indicated cells ($4 \times 10^6$) in a 100-mm culture dish were treated with 1% final concentration of formaldehyde to cross-link proteins to DNA, and the reaction was stopped by the addition of glycine. The cell lysates were sonicated to shear the DNA to fragments of 300–1,000 bp. Chromatin supernatants were incubated with anti-5mc (#15200081; Diagenode), anti-DNMT1 (#ab87656; Abcam), anti-DNMT3A (#ab2850; Abcam), anti-DNMT3B (#ab13604; Abcam), anti-H3K27me3 (#9733; Cell Signaling Technology), anti-H3K9me3 (#9753; Cell Signaling Technology), anti-H3K4me3 (#9751; Cell Signaling Technology), anti-SALL2 (#A303-208A; Bethyl), anti-p300 (#ab14984; Abcam), anti-RNA polymerase II (#05-623; Millipore), or anti-immunoglobulin G antibody (#I8765; Sigma-Aldrich) overnight at 4°C with rotation. After reversing the cross-linking of protein/DNA complexes to free DNA, PCR was performed using primers listed in Appendix Table S9.

## *In vivo* animal studies

Female BALB/c-nu mice (5 weeks old, 18–20 g) were purchased from the Slac-Jingda Animal Laboratory (Hunan, China) and housed in barrier facilities on a 12-h light/dark cycle. The mice were randomly divided into groups (*n* = 8/group). To investigate the effect of SALL2 on tamoxifen resistance in breast cancer, $5 \times 10^6$ MCF7 cells expressing either a SALL2 or control shRNA were injected into the mammary fat pads of female mice on the left side (control) and the right side (SALL2 shRNA) with 17β-estradiol pellets (E2, 0.72 mg/pellet, 60-day releasing, #SE-121; Innovative Research of America) 1 week before cancer cell implantation, and the tumor size was measured over time. Tamoxifen citrate (5 mg/pellet; 60-day release, #SE-351; Innovative Research of America) treatment was started 4 weeks after injection of MCF7 cells, when the tumors reached a volume of about 200 mm³. To explore the dependency of SALL2 knockdown cells on the PI3K/Akt pathway, approximately $5 \times 10^6$ cells were injected into the fat pads of the mice. After 4 weeks of injection of cells, Akt inhibitor ipatasertib (AZD5363) (#HY-15431; MedChemExpress) was administered orally for 6 weeks. To examine whether restoration of SALL2 in MCF7-TMR cells improved the anti-breast cancer effect of tamoxifen, approximately $5 \times 10^6$ cells were injected into the fat pads of the mice. When the xenografts reached a size of about 200 mm³ in 4 weeks, doxycycline (#D9891; Sigma-Aldrich) was

**The paper explained**

**Problem**

Tamoxifen is a first-line therapy and a widely used endocrine agent to treat ERα-positive breast cancer. Mortality of ERα-positive breast cancer patients is largely caused by the recurrence of tamoxifen-resistant tumors. Although loss of ERα is a crucial factor contributing to tamoxifen resistance, the molecular mechanism of ERα downregulation during tamoxifen therapy remains elusive. Further understanding of this mechanism is essential to develop novel strategies that will improve clinical outcomes for patients with endocrine-resistant breast cancer.

**Results**

Our study identified SALL2 as a key upstream factor of ERα in a subset of breast cancers. Promoter methylation-mediated loss of SALL2 conferred an estrogen-independent and tamoxifen-resistant phenotype to ERα-positive cancer cells via ERα downregulation and Akt/mTOR signaling activation. Importantly, *in vivo* experiments revealed that DNMT inhibitor-mediated SALL2 restoration resensitized tamoxifen-resistant breast cancer cells to tamoxifen therapy. Lower SALL2 expression or SALL2 hypermethylation predicted worse clinical outcomes in patients receiving tamoxifen therapy.

**Impact**

These results shed light on the mechanism of SALL2-mediated ERα regulation in a subset of breast cancers and highlight a clinical biosignature that could be used to identify breast cancer patients who might benefit from co-therapy with tamoxifen and DNMT inhibitor.

started in drinking water, or mice were treated with 5-Aza-dC (i.p.) for 4 days per cycle followed by 3 days of recovery, and the experimental animals were subcutaneously injected with or without a tamoxifen pellet. The kinetics of tumor formation were estimated by measuring the tumor volume every 3 days. Tumor volume was calculated using the equation (Length*Width²)/2. Six weeks later, tumors were excised and weighed. After the tumors were dissected, IHC analysis was performed using consecutive sections of tumors from three independent xenografted mice. The apoptotic index based on TUNEL staining (DeadEnd™ Fluorometric TUNEL System, #G3250; Promega) was quantified by measuring the percentage of TUNEL-positive cells. The proliferative index was quantified by measuring the percentage of Ki67-positive cells. All animal experimental procedures were approved by the Institutional Animal Care and Use Committee of Sun Yat-Sen University and performed in accordance with the relevant guidelines and regulations.

## Statistics

Statistical analyses were performed using GraphPad Prism 7 or SPSS 21.0. For comparison of two groups, *P*-values were calculated with a Student's *t*-test. For comparison of more than two groups, *P*-values were calculated using ANOVA test. The relationship between SALL2 expression and the clinicopathological characteristics was tested using the $\chi^2$ test. Survival curves were plotted with the Kaplan–Meier method and compared by the log-rank test. Multivariate survival analyses using Cox proportional hazard regression models were performed to evaluate independent

prognostic factors. $P < 0.05$ was considered statistically significant in all cases.

## Data availability

The RNA-seq data produced in this study have been deposited in the National Center for Biotechnology Information Sequence Read Archive (SRA) database with BioProject accession code PRJNA505938 (https://www.ncbi.nlm.nih.gov/bioproject/PRJNA505938/).

**Expanded View** for this article is available online.

## Acknowledgements

This work was supported by the National Natural Science Foundation of China (Nos. 91740118, 91529301, 81621004, 81530082, 81773106, 81830082, 91740119, 81872383, 81472546, 81672854, 81772826, 81802666, and 81802681); Natural Science Foundation of Guangdong Province (Nos. 2016A030308002, 2017A030306019, 2018B030311060, 2018B030311009, and 2018A030310321); Guangzhou Science and Technology Plan Projects (201803010098); Pearl River S&T Nova Program of Guangzhou (No. 201710010163); China Postdoctoral Science Foundation (No. 2017M622885); and the Fundamental Research Funds for the Central Universities (No. 17ykjc02).

## Author contributions

LY, CL, XWa, and QL designed the experiments and were responsible for all data collection and analysis. YL, MW, ZZ, and XWu conducted the bioinformatic analysis and animal experiments. DS, YX, LR, and YJ performed cellular experiments and biochemical experiments. MY, RO, and GD collected the clinical samples and conducted IHC analysis. YO and XC established the stable cell lines. JL and LS wrote the manuscript and supervised the project. All authors read and approved the final manuscript.

## Conflict of interest

The authors declare that they have no conflict of interest.

## For more information

(i)   TCGA: https://portal.gdc.cancer.gov/
(ii)  GSEA: http://software.broadinstitute.org/gsea/index.jsp
(iii) Kaplan–Meier plotter: http://kmplot.com/analysis/
(iv)  Cutoff Finder: http://molpath.charite.de/cutoff/index.jsp
(v)   the Human Protein Atlas (HPA) project: https://www.proteinatlas.org

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
