## [Review Process File · EMBO Molecular Medicine]

Epigenetic silencing of SALL2 confers tamoxifen resistance in breast cancer

Liping Ye, Chuyong Lin, Xi Wang, Qiji Li, Yue Li, Meng Wang, Zekun Zhao, Xianqiu Wu, Dongni Shi, Yunyun Xiao, Liangliang Ren, Yunting Jian, Meisongzhu Yang, Ruizhang Ou, Guangzheng Deng, Ying Ouyang, Xiangfu Chen, Jun Li & Libing Song

Review timeline:

Submission date:	26 March 2019
Editorial Decision:	3 May 2019
Revision received:	8 August 2019
Editorial Decision:	3 September 2019
Revision received:	19 September 2019
Accepted:	24 September 2019

Editor: Lise Roth

Transaction Report:

1st Editorial Decision

3 May 2019

Thank you for the submission of your manuscript to EMBO Molecular Medicine, and my apologies for the delay in getting back to you due to the late review of one referee. We have now heard back from the three referees whom we asked to evaluate your manuscript.

As you will see from the reports below, they are overall positive and support publication of the article in EMBO Molecular Medicine pending appropriate revisions. Particular attention should be given to strengthening the in vivo results, and to discussing the limitations of targeting SALL2 in the context of tamoxifen resistance. Addressing the reviewers' concerns in full will be necessary for further considering the manuscript in our journal. EMBO Molecular Medicine encourages a single round of revision only and therefore, acceptance or rejection of the manuscript will depend on the completeness of your responses included in the next, final version of the manuscript.

Please also contact us as soon as possible if similar work is published elsewhere. If other work is published, we may not be able to extend the revision period beyond three months.

I look forward to receiving your revised manuscript.

***** Reviewer's comments *****

Referee #1 (Comments on Novelty/Model System for Author):

SALL2 is not a novel target for the tamoxifen resistance and also it is very difficult to target SALL2 for the therapeutics.

Referee #1 (Remarks for Author):

The manuscript entitled "Epigenetic silencing of SALL2 confers tamoxifen resistance in breast cancer" by Ye et al., investigated the molecular mechanism of tamoxifen resistance in breast cancer and identified SALL2 plays an important role in tamoxifen resistance. The authors have used samples from the primary and tamoxifen resistant breast cancer tissues for the RNA-Seq analysis and identified several genes (165 genes) that are differentially expressed between primary and tamoxifen resistance. They have selected SALL2, although there is no convincing rational that is given to pick this gene, and characterized the role of this gene in tamoxifen resistance. They provided evidence that estrogen receptor (ER) is a direct target of SALL2 in tamoxifen resistance, and SALL2 reactivation resensitizes breast cancer cells for tamoxifen treatment, and the combination of SALL2 activator and tamoxifen would have better therapeutic option.

Although many molecular mechanisms have been identified for tamoxifen resistance, the authors have come up with a new molecular target for tamoxifen resistance. However, SALL2 is not a novel molecular target for tamoxifen resistance and also it is very difficult to target SALL2 for the therapeutic strategy. Because, SALL2 is regulated epigenetically, specifically DNA methylation, mediated gene silencing and DNA methyltransferase (DNMTs) inhibitors will reactivate SALL2 expression. However, DNMTs target many genes, DNMT inhibitors will reactivate several genes, including SALL2. There are many DNMT inhibitors in the clinics and the combination of DNMTs inhibitors and HDAC inhibitors has already been tested to overcome tamoxifen resistance. Although, DNMTs inhibitors showed very promising treatment option for the hematological malignancies, it has very limited benefits for solid cancers like breast cancer.

Overall the study is well designed and the experiments are well-articulated. Background information is very convincing and the results are well discussed. However, the overall results do not provide any new or novel information about the tamoxifen resistance.

Major points:

1. In figure 1, there are no details about the RNA samples that used for the RNA-Seq analysis. In the figure legend, they mentioned that 9 paired samples, from primary and tamoxifen resistance from the same patients, were used for the analysis. However, it is not clear whether they combined the primary and tamoxifen resistance samples together from 9 patients and carry out the analysis, because in Figure 1D, they labelled BC1 to BC9 and selected SALL2 from the list. Further, the heat map showed many genes like VWA2, ZG16B, and RARRES2 are more significantly altered than SALL2 but the authors have selected SALL2 but they did not provide any rational why they specifically selected SALL2.
2. Although the authors have claimed that SALL2 expression directly associated to ESR1 expression, they did not check the expression of SALL2 in ER-negative breast cancer tissues and or cell lines. Also, the authors have to test whether SALL2 expression in ER-negative breast cancer cells will reactivate ESR1 expression in these cell lines and will respond to tamoxifen treatment.

Overall the manuscript is not going to provide any novel mechanism for the tamoxifen resistance for the scientific community.

Referee #2 (Remarks for Author):

Ye et al report on the identification of SALL2 as a regulator of ESR1 and PTEN and its epigenetic silencing as a mechanism of tamoxifen resistance in breast cancer. Although aromatase inhibitors are increasingly used particularly in postmenopausal women, tamoxifen remains one of the

mainstays of endocrine therapy for hormone positive breast cancer. The paper is generally well written although could benefit from some mild editing for language and word choice. The work presented appears to be technically sound, although some clarifications are warranted in the description of methods. The study is quite impressive in its density and breadth of experiments and consistently supportive data to identify, characterize, and dissect the function of SALL2 and the consequence of SALL2 modulation in cancer cells and how this may be relevant to tamoxifen resistance. SALL2 appears to be a relatively strong direct regulator of both ESR1 and PTEN transcription through binding motifs in these genes' promoters. SALL2 hypermethylation appears to be relatively frequent in breast cancer and in patients treated with tamoxifen predicts both poor DFS and OS.

Specific Comments:

1. Data or references supporting the specificity of the SALL2 antibody for IHC should be provided. Is SALL2 found primarily in nuclei? Was staining of nuclei and membrane or cytoplasm scored the same? Also, was any optimization for the threshold between high and low SALL2 IHC performed?
2. What is the correlation between SALL2 hypermethylation and SALL2 IHC?
3. For the biomarker results to be translated into a possible tool for clinical use, it would be helpful to demonstrate that SALL2 methylation or protein level (IHC) predicts TAM resistance and that an alternative therapy (e.g. aromatase inhibitor) has greater efficacy for such patients. Do you have any data on how SALL2 modulates sensitivity to AIs, or for that matter, SERDS?
4. The methods for the RNA-sequencing is not adequately detailed. For example, it does not make sense that the total RNA was purified using a Nano Photometer. It is also not described how the mRNA was isolated from the RNA extracted from the FFPE sections, or was it perhaps from frozen tissue samples? The software used for gene expression estimated is not given. It would also be informative to indicate which metastatic site was sequenced (potential for organ-specific gene expression), and for both primary and metastasis, whether any gross- or micro-dissection was performed to select tumor-rich areas.
5. In the RNA-seq data, can you see expression of the alternative splice forms of SALL2 E1 and E1A? (see Hermosilla et al, Carcinogenesis 2017).
6. The discussion could benefit from a brief review of what is known about SALL2 and its potential role in cancer.

Minor Comments:

1. Figure 2E - "ER+SALL2_L" would be easier to interpret if written "ER+ SALL2_L".
2. Figure 7D and F - the legend is unclear as to what is represented by each bar. Also, the fold-change is from which endpoint compared to day 0? Is this data from the same experiments as in 7C and E?
3. Appendix Table S6 - the headers are unclear (Days to surgic, Days to treat tamoxifen, Days to relapse) and I would recommend not giving exact year/month/dates as this could be used to identify individuals, but rather give the same data in terms of number of days from a baseline (e.g. from day of diagnosis). It would also be informative to indicate which metastatic site was sequenced.
4. Figure 8 - spelling of "independent"
5. Page 8 - word choice is too strong in "To understand the intrinsic..." and "...which might be the underlying..." and "To explore the exact biological..."

Referee #3 (Comments on Novelty/Model System for Author):

In vivo studies need to be strengthened

Referee #3 (Remarks for Author):

Authors of this study examined the role of SALL2 gene in tamoxifen resistance in breast cancer. Authors show that SALL2 is epigenetically silenced in recurrent/metastatic breast cancer and DNMT inhibitors could restore the expression and sensitize tumors to tamoxifen in preclinical models. Authors also demonstrate that SALL2 and estrogen receptor expression are intimately connected through direct regulation of estrogen receptor expression by SALL2. Mechanistically, loss of SALL2 expression leads to reduced PTEN expression and consequently activation of

PI3K/AKT pathway. Authors suggest that activated PI3K/AKT pathway leads to estrogen-independent growth.

The majority of experiments are reasonably well performed and at least few experiments have been performed in more than one cell line. However, there are few instances where data are not completely supportive of their conclusions, particularly in vivo studies.

1) Authors claim that SALL2 knockdown results in estrogen -independent growth. However, all data are from in vitro studies. There is no evidence from in vivo experiments that cells with SALL2 knockdown can grow independent of estrogen pellet implantation. These experiments have not been performed. Dependency of SALL2 knockdown cells to PI3K/AKT pathway needs to be demonstrated in vivo using pathway inhibitors. Several of them are in clinic. Since Azacytidine has several targets, its in vivo activity in reversing tamoxifen resistance cannot be simply due to reactivation of SALL2. Only way to demonstrate that possibility is to show lack of effects of Azacytidine on MCF7-SALL2kd cells in vivo.

2) Binding of SALL2 to ESR1 regulatory regions was verified using overexpressed protein instead of endogenous SALL2. Since MCF-7 cells express significant levels of SALL2, it is surprising that authors had to overexpress SALL2 for CHIP assay. Similar concern with binding of SALL2 to PTEN regulatory regions. SALL2-dependent changes in p300, RNA pol II and AcH3 need to be determined in cells with and without SALL2 knockdown instead of overexpression studies. In addition, it is preferable to measure H3K4me3 rather than generic AcH3.

3) Few of the western blot results are not compatible with data in the literature. Tamoxifen usually does not cause degradation of ER, unlike data Figure 6G.

4) Tamoxifen resistance is often associated DNA-methylation dependent silencing of estrogen receptor target genes (Fan et al., Cancer Research 66:11954-66). Therefore, it is surprising to observe SALL2 overexpression in MCF7-TMR cells resulting in regain of tamoxifen sensitivity (Figure 7a and B). It is critical to evaluate the expression of several of estrogen-estrogen receptor target genes in these cells.

5) Authors need to authenticate all derivative cell lines to make sure the cell lines are MCF-7 cell derived (in case of MCF-7 derivatives) and not cross contaminating cell lines of the lab

6) Please specify sites of relapse as there is mounting evidence in the literature regarding metastasis site-specific genomic aberration.

7) Minor- please check spelling of methylation in Figure 6H

1st Revision - authors' response

8 August 2019

Reviewer's comments

Referee #1 (Comments on Novelty/Model System for Author):

SALL2 is not a novel target for the tamoxifen resistance and also it is very difficult to target SALL2 for the therapeutics.

Response: We are thankful to the reviewer for the comments. Although it has been well established that loss of ER expression is the dominant mechanism of tamoxifen resistance (1, 2), the reasons for the ER decrease during tamoxifen therapy remain largely unknown. Previously, several ESR1 regulators, such as Notch3, NR2E3 and MEL-18, have been reported to be involved in tamoxifen resistance (3-5). However, whether these regulators contribute to ER downregulation during tamoxifen therapy needed to be further investigated. To this end, we first profiled 9-paired pre-tamoxifen-treated primary breast cancer tissues and matched relapsed tamoxifen-resistant tissues from same individuals, and identified that transcription factor SALL2 was downregulated during tamoxifen therapy, which is not reported previously. Furthermore, by performing a serial of *in vitro* and *in vivo* experiments, we demonstrated that SALL2 acted as a key upstream regulator of ER expression via directly binding to ESR1 promoter. Importantly, doxycycline-inducible mouse model showed that restoration of SALL2 significantly increased the sensitivity of tamoxifen-resistant breast tumor to tamoxifen therapy. Therefore, these results suggested that SALL2 might be

a potential target in tamoxifen resistant breast cancer, which is unreported previously. However, at present, it still could not be achieved clinically to upregulation of SALL2 via transferring SALL2 cDNA into the breast tumor. We then further investigated the mechanism in which SALL2 was downregulated in breast cancer. Moreover, we found that hypermethylation status of SALL2 was significantly correlated with poorer clinical outcomes of tamoxifen-treated breast cancer patients. Restoration of SALL2 by DNMTs inhibitor 5-Aza-dC increased sensitivity of SALL2-hypermethylated ER-positive breast cancer cells, even of SALL2-hypermethylated/ESR1-hypomethylated ER-negative breast cancer cells, to tamoxifen therapy. However, 5-Aza-dC treatment did not result in ER restoration in SALL2-silenced tamoxifen-resistant cancer cells and only had marginal effect on growth inhibition. All these results suggested that targeting SALL2 via DNMTs inhibitor to reverse tamoxifen-resistance might be only benefit for breast cancer with SALL2 hypermethylation/ESR1-hypomethylation status. This is also consistent with clinical observation, as indicated by reviewer, that treatment with DNMTs inhibitor only would result in very limited benefits for all breast cancer patients. Therefore, our results not only shed light on the mechanism of SALL2 in reactivating ER expression, but also represent a potential clinical biosignature that used for subgrouping of breast cancer patients who might be benefit from co-therapy with tamoxifen and DNMTs inhibitor. The above-mentioned descriptions have been added in the Discussion section in the revised manuscript.

References

- (1) Osborne CK, et al. Mechanisms of endocrine resistance in breast cancer. *Annual review of medicine* 2011;62:233-47
- (2) Rondón-Lagos M, et al. Tamoxifen Resistance: Emerging Molecular Targets. *Int J Mol Sci.* 2016 Aug 19;17(8).
- (3) Dou, XW, et al. Notch3 Maintains Luminal Phenotype and Suppresses Tumorigenesis and Metastasis of Breast Cancer via Trans-Activating Estrogen Receptor- α . *Theranostics.* 2017; 7: 4041-4056
- (4) Park, YY, et al. Reconstruction of nuclear receptor network reveals that NR2E3 is a novel upstream regulator of ESR1 in breast cancer. *EMBO Mol Med.* 2012; 4: 52-67.
- (5) Lee, JY, et al. MEL-18 loss mediates estrogen receptor- α downregulation and hormone independence. *J Clin Invest.* 2015;125: 1801-1814.

Referee #1 (Remarks for Author):

The manuscript entitled "Epigenetic silencing of SALL2 confers tamoxifen resistance in breast cancer" by Ye et al., investigated the molecular mechanism of tamoxifen resistance in breast cancer and identified SALL2 plays an important role in tamoxifen resistance. The authors have used samples from the primary and tamoxifen resistant breast cancer tissues for the RNA-Seq analysis and identified several genes (165 genes) that are differentially expressed between primary and tamoxifen resistance. They have selected SALL2, although there is no convincing rationale that is given to pick this gene, and characterized the role of this gene in tamoxifen resistance. They provided evidence that estrogen receptor (ER) is a direct target of SALL2 in tamoxifen resistance, and SALL2 reactivation resensitizes breast cancer cells for tamoxifen treatment, and the combination of SALL2 activator and tamoxifen would have better therapeutic option.

Although many molecular mechanisms have been identified for tamoxifen resistance, the authors have come up with a new molecular target for tamoxifen resistance. However, SALL2 is not a novel molecular target for tamoxifen resistance and also it is very difficult to target SALL2 for the therapeutic strategy. Because, SALL2 is regulated epigenetically, specifically DNA methylation, mediated gene silencing and DNA methyltransferase (DNMTs) inhibitors will reactivate SALL2 expression. However, DNMTs target many genes, DNMT inhibitors will reactivate several genes, including SALL2. There are many DNMT inhibitors in the clinics and the combination of DNMTs inhibitors and HDAC inhibitors has already been tested to overcome tamoxifen resistance. Although, DNMTs inhibitors showed very promising treatment option for the hematological malignancies, it has very limited benefits for solid cancers like breast cancer.

Overall the study is well designed and the experiments are well-articulated. Background information is very convincing and the results are well discussed. However, the overall results do not provide any new or novel information about the tamoxifen resistance.

Major points:

1. In figure 1, there are no details about the RNA samples that used for the RNA-Seq analysis. In the figure legend, they mentioned that 9 paired samples, from primary and tamoxifen resistance from the same patients, were used for the analysis. However, it is not clear whether they combined the primary and tamoxifen resistance samples together from 9 patients and carry out the analysis, because in Figure 1D, they labelled BC1 to BC9 and selected SALL2 from the list. Further, the heat map showed many genes like VWA2, ZG16B, and RARRES2 are more significantly altered than SALL2 but the authors have selected SALL2 but they did not provide any rational why they specifically selected SALL2.

Response: We do appreciate for the reviewer's comments and we are sorry that we did not write these points clearly in the originally submitted manuscript. (1) The RNA-Seq assays were performed separately using 18 breast cancer samples, which comprising 9 pre-tamoxifen-treated primary breast cancer tissues and 9 matched relapsed tamoxifen-resistant tissues from the same individuals. The detailed information of these breast cancer tissues has been presented in Appendix Table S1. Then the RNA-Seq analysis was further performed to identify the dysregulated genes by comparing the relapsed tamoxifen-resistant tissues with the pre-tamoxifen-treated primary breast cancer tissues (As shown in Figure 1B-D). Meanwhile, inappropriate labeling in Figure 1D has been change to "Fold change (Rel./Pri.)", which each column represented patient-1 to patient-9.

(2) The rationale for SALL2 selection: Previously, it has been demonstrated that ER expression is a determinant of tamoxifen response in ER+ breast cancer, and loss of ER plays a vital role in tamoxifen resistance (1, 2). Notably, mRNA levels of ESR1 were found significantly decreased in a large cohort of tamoxifen-resistant breast cancer tissues (3-5), suggesting that transcriptional repression is involved in ESR1 downregulation. To further screen the key transcriptional factors that contribute to ESR1 downregulation, we then conducted RNA-seq assays in 9 paired pre-tamoxifen-treated primary breast cancer tissues and the matched relapsed tamoxifen-resistant tissues from same individuals. Consistently, RNA-seq analyses showed that ESR1 levels were significantly downregulated in 8/9 relapsed breast cancer tissues compared to the paired primary tumors (Appendix Figure S1A and B). We then screened the potential transcription factors involved in regulation of ESR1 and found that among 196 dysregulated genes, 50 genes expressions were significant correlated with ESR1 ($r > 0.6$) and only two transcription factors SALL2 and NKX3-1 were included (Figure 1B-D). However, silencing SALL2 significantly decreased ESR1 transcription levels and abolished the tamoxifen-induced growth inhibition in ER+ breast cancer cells, including MCF7, T47D, and ZR-75-1, but silencing NKX3-1 has no such effects (Appendix Figure S1C-F and Figure 1E). Moreover, statistical analysis revealed that lower SALL2 expression was significantly correlated with shorter relapse free, distant metastasis free and overall survival of ER+ breast cancer patients (Figure 1F). Therefore, SALL2 gene was selected for further study. The above-mentioned descriptions have been added in Results section in the revised manuscript.

References

- (1) Musgrove EA, et al. Biological determinants of endocrine resistance in breast cancer. *Nature reviews Cancer* 2009;9:631-43
- (2) Osborne CK, et al. Mechanisms of endocrine resistance in breast cancer. *Annual review of medicine* 2011;62:233-47
- (3) Kim C, et al. Estrogen receptor (ESR1) mRNA expression and benefit from tamoxifen in the treatment of estrogen receptor-positive breast cancer. *Journal of clinical oncology : official journal of the American Society of Clinical Oncology* 2011; 29:4160-7
- (4) Johnston SR, et al. Changes in estrogen receptor, progesterone receptor, and pS2 expression in tamoxifen-resistant human breast cancer. *Cancer research* 1995;55:3331-8
- (5) Drury SC, et al. Changes in breast cancer biomarkers in the IGF1R/PI3K pathway in recurrent breast cancer after tamoxifen treatment. *Endocrine-related cancer* 2011; 18:565-77

2. Although the authors have claimed that SALL2 expression directly associated to ESR1 expression, they did not check the expression of SALL2 in ER-negative breast cancer tissues and or cell lines. Also, the authors have to test whether SALL2 expression in ER-negative breast cancer cells will reactivate ESR1 expression in these cell lines and will respond to tamoxifen treatment.

Response: We thank the reviewer for the comments and the reviewer raised very interesting and important questions. As shown in Figure 2B, IHC analysis revealed that SALL2 protein expression was significantly reduced in ER-negative breast cancer tissues ($P < 0.001$; $n = 238$). Since ER promoter was found to be frequently methylated in ER-negative breast cancer tissues (1-2), which is one of mechanisms involved in loss of ER. Interestingly, we found that SALL2 was significantly associated with ER mRNA expression in ER-hypomethylated ER-negative breast cancer tissues ($P = 0.005$, $r = 0.420$; $n = 44$) but not in ER-hypermethylated ER-negative breast cancer tissues ($P =$

0.201, $r = 0.130$; $n = 99$) (Appendix Figure S8A). Using bisulfite PCR assays, we found that BT-549 and MDA-MB-231 ER-negative breast cancer cells were ER-hypermethylated but BT-20 ER-negative breast cancer cells was ER-hypomethylated (Appendix Figure S8B). Strikingly, overexpression of SALL2 significantly induced ESR1 mRNA expression in BT-20 cells but had no effect on the ESR1 mRNA expression in BT-549 and MDA-MB-231 cells (Appendix Figure S8C). Moreover, we found that overexpressing SALL2 dramatically increased the tamoxifen response in BT-20 cells but not in BT-549 and MDA-MB-231 cells (Appendix Figure S8D). These results provided further evidence that SALL2 could reactivate ER expression and enhance tamoxifen response in breast cancer. The above-mentioned results have been incorporated into the revised manuscript.

References

- (1) Lapidus RG, et al. Mapping of ER gene CpG island methylation-specific polymerase chain reaction. *Cancer research* 1998;58:2515-9
- (2) Yang X, et al. Synergistic activation of functional estrogen receptor (ER)-alpha by DNA methyltransferase and histone deacetylase inhibition in human ER-alpha-negative breast cancer cells. *Cancer research* 2001;61:7025-9

Overall the manuscript is not going to provide any novel mechanism for tamoxifen resistance for the scientific community.

Response: We are thankful to the reviewer for the comment. Although loss of ER expression is the dominant mechanism of tamoxifen resistance (1, 2), the reasons for the ER decrease during tamoxifen therapy remain largely unknown. Previously, several ESR1 regulators, such as Notch3, NR2E3 and MEL-18, have been reported to be involved in tamoxifen resistance (3-5). However, whether these regulators contribute to ER downregulation during tamoxifen therapy needed to be further investigated. To this end, we first profiled 9-paired pre-tamoxifen-treated primary breast cancer tissues and matched relapsed tamoxifen-resistant tissues from the same individuals, and identified that transcription factor SALL2 was downregulated during tamoxifen therapy, which is not reported previously. Furthermore, by performing a serial of *in vitro* and *in vivo* experiments, we demonstrated that SALL2 acted as a key upstream regulator of ER expression via directly binding to ESR1 promoter. Importantly, doxycycline-inducible mouse model showed that restoration of SALL2 significantly increased the sensitivity of tamoxifen-resistant breast tumor to tamoxifen therapy. Therefore, these results suggested that SALL2 might be a potential target in tamoxifen resistant breast cancer, which is unreported previously. However, at present, it still could not be achieved clinically to upregulation of SALL2 via transferring SALL2 cDNA into the breast tumor. We then further investigated the mechanism in which SALL2 was downregulated in breast cancer. Moreover, we found that hypermethylation status of SALL2 was significantly correlated with poorer clinical outcomes of tamoxifen-treated breast cancer patients. Restoration of SALL2 by DNMTs inhibitor 5-Aza-dC increased sensitivity of SALL2-hypermethylated ER-positive breast cancer cells, even SALL2-hypermethylated/ESR1-hypomethylated ER-negative breast cancer cells, to tamoxifen therapy. However, 5-Aza-dC treatment did not result in ER restoration in SALL2-silenced tamoxifen-resistant cancer cells and only had marginal effect on growth inhibition. All these results suggested that targeting SALL2 via DNMTs inhibitor to reverse tamoxifen-resistance might be only benefit for breast cancer with SALL2 hypermethylation/ESR1-hypomethylation status. This is also consistent with clinical observation, as indicated by reviewer, that treatment with DNMTs inhibitor only would result in very limited benefits for all breast cancer patients. Therefore, our results not only shed light on the mechanism of SALL2 in reactivating ER expression, but also represent a potential clinical biosignature that used for subgrouping of breast cancer patients who might be benefit from co-therapy with tamoxifen and DNMTs inhibitor. The above-mentioned descriptions have been added in the Discussion section in the revised manuscript.

References

- (1) Osborne CK, et al. Mechanisms of endocrine resistance in breast cancer. *Annual review of medicine* 2011;62:233-47
- (2) Rondón-Lagos M, et al. Tamoxifen Resistance: Emerging Molecular Targets. *Int J Mol Sci.* 2016 Aug 19;17(8).
- (3) Dou, XW, et al. Notch3 Maintains Luminal Phenotype and Suppresses Tumorigenesis and Metastasis of Breast Cancer via Trans-Activating Estrogen Receptor-alpha. *Theranostics.* 2017; 7: 4041-4056
- (4) Park, YY, et al. Reconstruction of nuclear receptor network reveals that NR2E3 is a novel upstream regulator of ESR1 in breast cancer. *EMBO Mol Med.* 2012; 4: 52-67.
- (5) Lee, JY, et al. MEL-18 loss mediates estrogen receptor-alpha downregulation and hormone

independence. *J Clin Invest.* 2015;125: 1801-1814.

Referee #2 (Remarks for Author):

Ye et al report on the identification of SALL2 as a regulator of ESR1 and PTEN and its epigenetic silencing as a mechanism of tamoxifen resistance in breast cancer. Although aromatase inhibitors are increasingly used particularly in postmenopausal women, tamoxifen remains one of the mainstays of endocrine therapy for hormone positive breast cancer. The paper is generally well written although could benefit from some mild editing for language and word choice. The work presented appears to be technically sound, although some clarifications are warranted in the description of methods. **The study is quite impressive in its density and breadth of experiments and consistently supportive data to identify, characterize, and dissect the function of SALL2 and the consequence of SALL2 modulation in cancer cells and how this may be relevant to tamoxifen resistance.** SALL2 appears to be a relatively strong direct regulator of both ESR1 and PTEN transcription through binding motifs in these genes' promoters. SALL2 hypermethylation appears to be relatively frequent in breast cancer and in patients treated with tamoxifen predicts both poor DFS and OS.

Specific Comments:

1. Data or references supporting the specificity of the SALL2 antibody for IHC should be provided. Is SALL2 found primarily in nuclei? Was staining of nuclei and membrane or cytoplasm scored the same? Also, was any optimization for the threshold between high and low SALL2 IHC performed?

Response: We do appreciate the reviewer for raising these important points. The IHC specificity of anti-SALL2 antibody (Sigma, HPA004162), which was used in current study for IHC analysis of SALL2 protein, has been validated by the Human Protein Atlas (HPA) project (<https://www.proteinatlas.org/ENSG00000165821-SALL2/antibody>). To further confirm the specificity of anti-SALL2 antibody in breast cancer tissues, the primary antibody was replaced with anti-IgG antibody, or blocked with human recombinant SALL2 protein by co-incubation at 4°C overnight preceding the immunohistochemical staining procedure. As shown in Appendix Figure S4, we did not find positive SALL2 signals in paraffin-embedded breast cancer samples using anti-SALL2 antibody co-incubated with recombinant SALL2 protein, similar to anti-IgG antibody. Therefore, these results provided further evidence that anti-SALL2 antibody used for IHC staining in breast cancer samples was specific.

In the current study, IHC analysis was performed on the 238 paraffin-embedded breast cancer tissues using the anti-SALL2 rabbit polyclonal antibody (1:100 dilution; #HPA004162, Sigma-Aldrich). As shown in Figure 2A and Appendix Figure S4, IHC analysis of SALL2 in breast cancer tissues displayed that SALL2 protein was primarily localized in nuclei and only showed marginal cytoplasmic expression, which was consistent with previous reports (1). The IHC staining was then scored separately by two independent pathologists. Tumor cell proportions were scored as follows: 0, no positive tumor cells; 1, <10% positive tumor cells; 2, 10–35% positive tumor cells; 3, 35–75% positive tumor cells; 4, >75% positive tumor cells. Staining intensity was graded according to the following standard: 0, no staining; 1, weak staining (light yellow); 2, moderate staining (dark yellow/light brown); 3, strong staining (dark brown). The staining index (SI) was calculated as the product of the staining intensity score and the proportion of positive tumor cells. An optimal threshold of $SI \geq 6$ was then determined to define tumors with high expression using the Cutoff Finder program (<http://molpath.charite.de/cutoff/>) (2). Appropriate descriptions have been added in Methods section in the revised manuscript.

References

- (1) Nielsen TO, et al. Tissue microarray validation of epidermal growth factor receptor and SALL2 in synovial sarcoma with comparison to tumors of similar histology. *Am J Pathol.* 2003; 163(4):1449-56.
- (2) Budczies, J, et al. Cutoff Finder: a comprehensive and straightforward Web application enabling rapid biomarker cutoff optimization. *PLoS One* 2012; 7: e51862.

2. What is the correlation between SALL2 hypermethylation and SALL2 IHC?

Response: We thank the reviewer for the comment. As shown in Figure 6G, SALL2-hypermethylation status was significantly associated with low expression of SALL2 analyzed by IHC staining in human breast cancer specimens ($P < 0.001$; $n = 90$). The above-mentioned results have been incorporated into the revised manuscript.

3. For the biomarker results to be translated into a possible tool for clinical use, it would be helpful to demonstrate that SALL2 methylation or protein level (IHC) predicts TAM resistance and that an alternative therapy (e.g. aromatase inhibitor) has greater efficacy for such patients. Do you have any data on how SALL2 modulates sensitivity to AIs, or for that matter, SERDS?

Response: We thank the reviewer for the comments and the reviewer raised an important and interesting suggestion for clinical translation of our results. In the present study, we found that either hypermethylation status of SALL2 or lower expression of SALL2 protein was significantly correlated with shorter disease free survival and overall survival of tamoxifen-treated breast cancer patients (Figure 2G and 6H), suggesting that the SALL2 methylation status or SALL2 low expression may be a potential marker for the prediction of tamoxifen therapy resistance. We currently do not have enough breast cancer samples treated with AIs or SERDS. However, for this important and interesting suggestion, we would collect more AIs- or SERDS-treated breast cancer samples to further examine correlation between SALL2 methylation status and SALL2 low expression with outcome of AIs or SERDS therapy, which would provide a potential clinical biomarker for an alternative therapy of breast cancer. The above-mentioned descriptions have been added in Discussion section in the revised manuscript.

4. The methods for the RNA-sequencing is not adequately detailed. For example, it does not make sense that the total RNA was purified using a Nano Photometer. It is also not described how the mRNA was isolated from the RNA extracted from the FFPE sections, or was it perhaps from frozen tissue samples? The software used for gene expression estimated is not given. It would also be informative to indicate which metastatic site was sequenced (potential for organ-specific gene expression), and for both primary and metastasis, whether any gross- or micro-dissection was performed to select tumor-rich areas.

Response: We thank for the reviewer's comments. We apologized for the wrong description, "total RNA was purified using a Nano Photometer" should be "The RNA purity was evaluated using a Nano Photometer® spectrophotometer (IMPLEN, Westlake Village, CA, USA)", which has been corrected in the revised manuscript. The frozen breast cancer samples used for RNA-seq were obtained from primary tumor tissues, at diagnosis before tamoxifen treatment, and from relapsed tumor after tamoxifen from the same individuals. The information of relapse sites, including bone (BN), brain (BR), chest wall (CW), lung (LU), liver (LI), and ovary (OV), has been added in Appendix Table S1 in the revised manuscript. The gross-dissection was performed to select tumor-rich areas. The above-mentioned descriptions have been added in Methods section in the revised manuscript.

We are sorry that we did not write this information clearly in the originally submitted manuscript. The detailed information of RNA-sequencing has been added in Methods section in the revised manuscript, as following: the total RNA was extracted from the frozen breast cancer tissues using the TRIzol reagent (#15596018, Invitrogen; Life Technologies, USA) according to the manufacturer's instructions. After cDNA library was generated and sequenced, the clean reads with initial quality control were mapped to human genome (hg38) using TopHat2 with a reference annotation from the Ensembl database (1). A gene expression profile of each sample, were quantitatively analyzed using the Cuffdiff suite from Cufflinks (v2.2.1) (2), which is a program for the comparative assembly of transcripts and the estimation of their abundance in a transcriptome sequencing experiment by using the measurement unit FPKM (fragments per kilobase of transcript per million mapped reads).

References

- (1) Kim, D, et al. TopHat2: accurate alignment of transcriptomes in the presence of insertions, deletions and gene fusions. *Genome Biol* 2013; 14: R36.
- (2) Trapnell, C, et al. Differential gene and transcript expression analysis of RNA-seq experiments with TopHat and Cufflinks. *Nat Protoc* 2012; 7: 562-578.

5. In the RNA-seq data, can you see expression of the alternative splice forms of SALL2 E1 and E1A? (see Hermosilla et al, *Carcinogenesis* 2017).

Response: We thank for the reviewer's comment. As suggested by the reviewer, we re-checked our RNA-seq data and found that only E1A isoform was expressed in our collected breast cancer samples, which is further confirmed by qPCR analysis (Appendix Figure S3A). Consistently, TCGA analysis also revealed that only E1A isoform but not E1 isoform of SALL2 was expressed in all breast cancer tissues (Appendix Figure S3B). Since it has been reported that SALL2 E1 isoform transcript is only restricted to certain tissues, such as thymus and testis (1-2). These results indicate

that the E1A isoform of SALL2 plays an important role in breast cancer. The above-mentioned descriptions have been added in the revised manuscript.

References

- (1) Hermosilla VE, et al. Developmental SALL2 transcription factor: a new player in cancer. *Carcinogenesis* 2017;38:680-90
- (2) Ma, Y, et al. Cloning and characterization of two promoters for the human HSAL2 gene and their transcriptional repression by the Wilms tumor suppressor gene product. *J Biol Chem* 2001; 276: 48223-48230.

6. The discussion could benefit from a brief review of what is known about SALL2 and its potential role in cancer.

Response: We are thankful for the reviewer's comment and the reviewer's point is well taken. The brief review regarding of previous studies on SALL2 and its potential role in cancer has been discussed in the revised manuscript, as following: SALL2, a member of spalt like transcription factor family, plays important roles in various biological processes, such as neurogenesis, neuronal differentiation, kidney development and eye morphogenesis (1-3). It has been reported that SALL2 contributes to cellular apoptosis, growth arrest and quiescence maintenance by regulation of multiple downstream genes, such as p21^{Cip1/Waf1}, p16^{Ink4a}, c-MYC, BAX and Noxa (4-8). Recently, downregulation of SALL2 was found in various types of cancer, including leukemia, ovarian, lung, and radioresistant esophageal cancers, and involved in cancer development and progression (9-11), suggesting that SALL2 is also involved in human cancers. Furthermore, it has been reported that SALL2 bound with polyomavirus large T antigen and papilloma virus E6 protein and inhibited their functions to promote tumor development (12, 13). Double knockout of Sall2 and p53 in mice (Sall2^{-/-}/p53^{-/-}) accelerated tumorigenesis, tumor progression, and causes significantly higher mortality and metastasis rates compared with p53^{-/-} mice (14). Moreover, immortalized Sall2^{-/-} mouse embryonic fibroblasts (MEFs) showed enhanced growth rate, foci formation, and anchorage-independent growth, indicating that SALL2 might be a tumor suppressor (15). Consistently, in the present study, we found that low expression of SALL2 correlated with relapse and poor survival of breast cancer patient. Subsequently, we demonstrated that loss of SALL2 promoted tumorigenicity of breast cancer cells *in vitro* and *in vivo*, while overexpression of SALL2 had opposite effects. Thus, our results further support an important tumor suppressive role of SALL2 in human cancers.

References

- (1) Kelberman D, et al. Mutation of SALL2 causes recessive ocular coloboma in humans and mice. *Human molecular genetics* 2014;23:2511-26
- (2) Chatterjee R, et al. Targeted exome sequencing integrated with clinicopathological information reveals novel and rare mutations in atypical, suspected and unknown cases of Alport syndrome or proteinuria. *PloS one* 2013;8:e76360
- (3) Bohm J, et al. Sall1, sall2, and sall4 are required for neural tube closure in mice. *The American journal of pathology* 2008;173:1455-63
- (4) Dawei Li, et al. p150 (Sal2) is a p53-independent regulator of p21 (WAF1/CIP). *Mol Cell Biol* 2004
- (5) Wu Z, et al. Sal-like protein 2 upregulates p16 expression through a proximal promoter element. *Cancer science* 2015;106:253-61
- (6) Sung CK, et al. The polyoma virus large T binding protein p150 is a transcriptional repressor of c-MYC. *PloS one* 2012;7:e46486
- (7) Gu H, et al. DNA-binding and regulatory properties of the transcription factor and putative tumor suppressor p150 (Sal2). *Biochimica et biophysica acta* 2011;1809:276-83
- (8) Escobar D, et al. Sall2 is required for proapoptotic Noxa expression and genotoxic stress-induced apoptosis by doxorubicin. *Cell Death Dis* 2015;6:e1816
- (9) Sung CK, et al. Promoter methylation of the SALL2 tumor suppressor gene in ovarian cancers. *Mol Oncol* 2013;7:419-27
- (10) Luo J, et al. mRNA and methylation profiling of radioresistant esophageal cancer cells: the involvement of Sall2 in acquired aggressive phenotypes. *J Cancer* 2017;8:646-56
- (11) Liu LY, et al. A supervised network analysis on gene expression profiles of breast tumors predicts a 41-gene prognostic signature of the transcription factor MYB across molecular subtypes. *Comput Math Methods Med* 2014;2014:813067
- (12) Li D, et al. A tumor host range selection procedure identifies p150 (sal2) as a target of polyoma virus large T antigen. *Proc Natl Acad Sci U S A* 2001;98:14619-24
- (13) Parroche P, et al. Human papillomavirus type 16 E6 inhibits p21 (WAF1) transcription

- independently of p53 by inactivating p150 (Sal2). *Virology* 2011;417:443-8
- (14) Chai L. The role of HSAL (SALL) genes in proliferation and differentiation in normal hematopoiesis and leukemogenesis. *Transfusion* 2011;51 Suppl 4:87S-93S
- (15) V EH, et al. SALL2 represses cyclins D1 and E1 expression and restrains G1/S cell cycle transition and cancer-related phenotypes. *Mol Oncol* 2018;12:1026-46

Minor Comments:

1. Figure 2E - "ER+SALL2_L" would be easier to interpret if written "ER+ SALL2_L".

Response: The reviewer's point is well-taken. We are sorry for the confusion that "ER+SALL2_L" means ER- breast cancer with SALL2 lower expression. Appropriate modification has been made in Figure 2E in the revised manuscript, which "ER+SALL2_L" has been changed to "ER-SALL2_L".

2. Figure 7D and F - the legend is unclear as to what is represented by each bar. Also, the fold-change is from which endpoint compared to day 0? Is this data from the same experiments as in 7C and E?

Response: The reviewer's points are well-taken and we are sorry that we did not write this point clearly in the initial submitted manuscript. The data showed in Figure 7D and F were from the same experiments as shown in Figure 7C and E. Each bar in Figure 7D and F represented the fold change of tumor volume that was calculated as $(V_t - V_i) / V_i$, where V_t represents the final tumor volume at the end of drug treatment, and V_i stands for the initial tumor volume starting treatment. As the results in Figure 7D and F were another displaying manner of Figure 7C and E, which may cause confusion to the readers, we therefore deleted the original Figure 7D and F in the revised manuscript.

3. Appendix Table S6 - the headers are unclear (Days to surgic, Days to treat tamoxifen, Days to relapse) and I would recommend not giving exact year/month/dates as this could be used to identify individuals, but rather give the same data in terms of number of days from a baseline (e.g. from day of diagnosis). It would also be informative to indicate which metastatic site was sequenced.

Response: We do appreciate the reviewer's comments and the reviewer's point is well taken. As suggested by the reviewer, the number of days from tamoxifen treatment to relapse and the specific metastatic site sequenced have been adopted in new Appendix Table S1 in the revised manuscript.

4. Figure 8 - spelling of "independent"

Response: We apologize for the typing error and thank the reviewer for pointing it out. Appropriate correction has been made in Figure 8 in the revised manuscript.

5. Page 8 - word choice is too strong in "To understand the intrinsic..." and "...which might be the underlying..." and "To explore the exact biological..."

Response: We are thankful for the reviewer's comment and the reviewer's point is well taken. As suggested by the reviewer, all these inappropriate words, such as "intrinsic", "underlying" and "exact", have been deleted in the revised manuscript.

Referee #3 (Comments on Novelty/Model System for Author):

In vivo studies need to be strengthened

Response: We thank the reviewer for the comments and these points are well taken. As requested by the reviewer, multiple *in vivo* experiments have been performed to provide more evidence for further strengthening our conclusion. We first examined the effect of downregulation of SALL2 on estrogen-independent tumor formation using *in vivo* model. As shown in Figure 4D and E, the MCF7/SALL2-Ri#1 cells could form tumors in the mammary fat pad of nude mice after one month of inoculation without estrogen pellet implantation, whereas MCF7/control cells failed to form tumors in the fat pad of mice without estrogen treatment. These results further supported the notion that silencing SALL2 conferred estrogen-independent tumorigenicity of ER+ breast cancer cells.

Furthermore, the crucial effect of PI3K/Akt signaling on SALL2 downregulation-induced tumorigenicity of ER+ breast cancer cells was assessed. As shown in Figure 5E-G, administration of Ipatasertib (GDC-0068), an Akt inhibitor that is used in clinical breast cancer therapy (1-2), significantly inhibited the tumorigenic capability of MCF7/SALL2-Ri#1 cells upon estrogen treatment. More importantly, Ipatasertib treatment completely abrogated the tumor formation by MCF7/SALL2-Ri#1 cells in the fat pad of nude mice without estrogen pellet implantation (Figure

5E and F). Thus, these results indicated that PI3K/Akt pathway was required for SALL2 knockdown-induced tumor growth independent of estrogen.

Moreover, we further examined the *in vivo* anti-tumor activity of 5-aza-2'-deoxycytidine (5-Aza-dC), a deoxy derivative of azacytidine, using MCF7-TMR/SALL2-Ri#1 and MCF7/SALL2-Ri#1 cells. As shown in Figure 7F-H and Appendix Figure S9C-E, administration of 5-Aza-dC did not abrogate the tumorigenic capability of MCF7-TMR/SALL2-Ri#1 and MCF7/SALL2-Ri#1 cells with or without tamoxifen treatment, which provided further evidence that SALL2 plays a vital role in 5-Aza-dC-induced tamoxifen sensitization.

All these above-mentioned results have been incorporated into the revised manuscript.

References

- (1) Lin J, et al. Targeting activated Akt with GDC-0068, a novel selective Akt inhibitor that is efficacious in multiple tumor models. *Clin Cancer Res.* 2013; 19:1760–1772.
- (2) Kim SB, et al. Ipatasertib plus paclitaxel versus placebo plus paclitaxel as first-line therapy for metastatic triple-negative breast cancer (LOTUS): a multicentre, randomised, double-blind, placebo-controlled, phase 2 trial. *Lancet Oncol.* 2017; 18(10):1360-1372.

Referee #3 (Remarks for Author):

Authors of this study examined the role of SALL2 gene in tamoxifen resistance in breast cancer. Authors show that SALL2 is epigenetically silenced in recurrent/metastatic breast cancer and DNMT inhibitors could restore the expression and sensitize tumors to tamoxifen in preclinical models. Authors also demonstrate that SALL2 and estrogen receptor expression are intimately connected through direct regulation of estrogen receptor expression by SALL2. Mechanistically, loss of SALL2 expression leads to reduced PTEN expression and consequently activation of PI3K/AKT pathway. Authors suggest that activated PI3K/AKT pathway leads to estrogen-independent growth.

The majority of experiments are reasonably well performed and at least few experiments have been performed in more than one cell line. However, there are few instances where data are not completely supportive of their conclusions, particularly *in vivo* studies.

1. Authors claim that SALL2 knockdown results in estrogen-independent growth. However, all data are from *in vitro* studies. There is no evidence from *in vivo* experiments that cells with SALL2 knockdown can grow independent of estrogen pellet implantation. These experiments have not been performed. Dependency of SALL2 knockdown cells to PI3K/AKT pathway needs to be demonstrated *in vivo* using pathway inhibitors. Several of them are in clinic. Since Azacytidine has several targets, its *in vivo* activity in reversing tamoxifen resistance cannot be simply due to reactivation of SALL2. Only way to demonstrate that possibility is to show lack of effects of Azacytidine on MCF7-SALL2kd cells *in vivo*.

Response: We thank the reviewer for the comments and these points are well taken. As requested by the reviewer, multiple *in vivo* experiments have been performed to provide more evidence for further strengthening our conclusion. We first examined the effect of SALL2 downregulation on estrogen-independent tumor formation in *in vivo* model. As shown in Figure 4D and E, the MCF7/SALL2-Ri#1 cells could form tumors in the mammary fat pad of nude mice after one month of inoculation without estrogen pellet implantation, whereas MCF7/control cells failed to form tumors in the fat pad of mice without estrogen treatment. These results further supported the notion that silencing SALL2 conferred estrogen-independent tumorigenicity of human ER+ breast cancer cells. *In vivo* model was further employed to examine the crucial effect of PI3K/AKT signaling on SALL2 downregulation-induced tumorigenicity of ER+ breast cancer cells. As shown in Figure 5E-G, administration of Ipatasertib (GDC-0068), an Akt inhibitor used in clinical breast cancer therapy (1-2), significantly inhibited the tumor formation capability of MCF7/SALL2-Ri#1 cells upon estrogen treatment. Importantly, Ipatasertib treatment almost completely abolished the tumorigenic capability of MCF7/SALL2-Ri#1 cells in the fat pad of nude mice without estrogen pellet implantation (Figure 5E and F). Thus, these results indicated that PI3K/AKT pathway was required for SALL2 silencing-induced tumor growth independent of estrogen treatment. Moreover, we examined the *in vivo* anti-tumor activity of 5-aza-2'-deoxycytidine (5-Aza-dC) in SALL2-silenced cells. As shown in Figure 7F-H and Appendix Figure S9C-E, 5-Aza-dC treatment failed to abrogate the tumorigenic capability of SALL2-silenced-MCF7/TMR and -MCF7 cells with or without tamoxifen treatment, which provided further evidence that SALL2 is required for 5-Aza-dC-induced tamoxifen sensitization. All above-mentioned results have been incorporated into the revised manuscript.

References

- (1) Lin J, et al. Targeting activated Akt with GDC-0068, a novel selective Akt inhibitor that is efficacious in multiple tumor models. *Clin Cancer Res.* 2013; 19:1760–1772.
- (2) Kim SB, et al. Ipatasertib plus paclitaxel versus placebo plus paclitaxel as first-line therapy for metastatic triple-negative breast cancer (LOTUS): a multicentre, randomised, double-blind, placebo-controlled, phase 2 trial. *Lancet Oncol.* 2017; 18(10):1360-1372.

2. Binding of SALL2 to ESR1 regulatory regions was verified using overexpressed protein instead of endogenous SALL2. Since MCF7 cells express significant levels of SALL2, it is surprising that authors had to overexpress SALL2 for CHIP assay. Similar concern with binding of SALL2 to PTEN regulatory regions. SALL2-dependent changes in p300, RNA pol II and AcH3 need to be determined in cells with and without SALL2 knockdown instead of overexpression studies. In addition, it is preferable to measure H3K4me3 rather than generic AcH3.

Response: We do appreciate the reviewer's comment and the reviewer's point is well taken. As requested by the reviewer, we further performed the ChIP assays in MCF7 cells using the anti-SALL2 antibody (#A303-208A, Bethyl) to examine the interactions of endogenous SALL2 with its target gene promoters. As shown in Figure 3D and Appendix Figure S7D, ChIP assays revealed that endogenous SALL2 protein could also interact with the promoter of *ESR1* and *PTEN*. Meanwhile, we found that the enrichment of p300, RNA pol II and H3K4me3 on the *ESR1* and *PTEN* promoter was significantly reduced in SALL2-silenced MCF-7 cells compared to control cells (Figure 3E and Appendix Figure S7E). These results provided the further evidence that SALL2 plays an important role in regulation of the promoter activity of ESR1 and PTEN in breast cancer cells. The above-mentioned results have been incorporated into the revised manuscript.

3. Few of the western blot results are not compatible with data in the literature. Tamoxifen usually does not cause degradation of ER, unlike data Figure 6G.

Response: We thank the reviewer for the comment and we are sorry that we did not write this point clearly in originally submitted version. The expression of ER in Figure 6G, which is the Figure 7E in the current version, was examined in MCF7-TMR/tumors upon tamoxifen treatment for 6 weeks but not in MCF7-TMR cells. As indicated by the reviewer, we also did not observe the ER downregulation in tamoxifen-treated MCF7, MCF7-TMR and ZR-75-30 cells (Appendix Figure S10, for reviewer's and editor's reference), which is consistent with previous reports. Interestingly, we found that SALL2 expression was also reduced in tamoxifen-treated MCF7-TMR/tumors, which suggested that the decreased ER in tamoxifen-treated MCF7-TMR/tumors might be due to SALL2 downregulation, as administration of 5-Aza-dC-induced SALL2 restoration was accompanied with recovering of ER level in tamoxifen-treated MCF7-TMR/tumors (Figure 7E). We hypothesized that the tumor microenvironment and intracellular biological stimuli, such as the hypoxia, acidic pH and elevated reductive conditions within organelles, might be involved in SALL2 reduction-mediated ER downregulation in breast tumor. Therefore, these results further support the notion that SALL2 contributes to regulation of ER in breast cancer tumors.

4. Tamoxifen resistance is often associated DNA-methylation dependent silencing of estrogen receptor target genes (Fan et al., *Cancer Research* 66:11954-66). Therefore, it is surprising to observe SALL2 overexpression in MCF7-TMR cells resulting in regain of tamoxifen sensitivity (Figure 7A and B). It is critical to evaluate the expression of several of estrogen-estrogen receptor target genes in these cells.

Response: We thank the reviewer for the comments and the reviewer raised an important point. Previously, Fan M, et al. reported that approximately 40% of estrogen downstream genes were significantly changed, but they only found two ER downstream genes, including ID4 and FABP5, exhibited significant alterations in promoter methylation, in tamoxifen resistant MCF7 (MCF7-T) cells compared to MCF7/control cells (1), suggesting that the alterations of most estrogen receptor target genes in MCF7-T cells were not due to DNA-methylation-mediated silencing. Interestingly, our results revealed that SALL2 overexpression in MCF7-TMR cells resulted in significant upregulation of multiple ER target genes, including CA12, RET, STC2, KAAG1, PMEPA1, MAST4, MSX2 and GFRA1, which provided further evidence that SALL2 plays a critical role in regain of tamoxifen sensitivity in breast cancer (Figure 4G). The above-mentioned results have been incorporated into the revised manuscript.

Reference

- (1) Fan M, et al. Diverse gene expression and DNA methylation profiles correlate with differential adaptation of breast cancer cells to the antiestrogens tamoxifen and fulvestrant. *Cancer*

research 2006;66:11954-66

5. Authors need to authenticate all derivative cell lines to make sure the cell lines are MCF7 cell derived (in case of MCF7 derivatives) and not cross contaminating cell lines of the lab

Response: The reviewer's point is well taken. As suggested by the reviewer, authentication has been performed by short tandem repeat (STR) fingerprinting at Medicine Lab of Forensic Medicine Department of Sun Yat-Sen University (China). The authentication showed that MCF7-TMR and parental MCF7 have the same STR features (D5S818: 12; D13S317: 11; D7S820: 8, 9; D16S539: 11, 12; VWA: 14, 15; TH01: 6; AMEL: X; TPOX: 9, 12; CSF1PO: 10), which indicated that MCF7-TMR is a derivative of MCF7.

6. Please specify sites of relapse as there is mounting evidence in the literature regarding metastasis site specific genomic aberration.

Response: We thank for the reviewer's comment and we are sorry that we did not write this information clearly. The information of specify relapse sites has been added in Appendix Table S1 in the revised manuscript.

7. Minor- please check spelling of methylation in Figure 6H

Response: We apologize for the typing error and thank the reviewer for pointing it out. Appropriate correction has been made in Figure 6H in the revised manuscript.

2nd Editorial Decision

3 September 2019

Thank you for the submission of your revised manuscript to EMBO Molecular Medicine. We have now received the referees' reports, and as you will see they are supportive of publication of your study pending minor revisions. I am therefore pleased to inform you that we will be able to accept your manuscript, once the following concerns will be addressed:

1) Please address the comments from referee #1. At this stage, we'd like you to discuss the referee's points in writing, and add the additional controls/quantifications requested. We do not ask you for any far-reaching experiment.

I look forward to reading a new revised version of your manuscript as soon as possible.

***** Reviewer's comments *****

Referee #1 (Comments on Novelty/Model System for Author):

The Overall impact of the manuscript is not novel and the experimental technics also not novel but adequate to prove the central hypothesis.

Referee #1 (Remarks for Author):

In this revised manuscript, the authors have successfully addressed most of the concerns raised by all the reviewers. They have performed several new experiments and provided very logical explanation for most of the critics raised by the reviewers. However, still some concerns or issues in the manuscripts:

1) As mentioned in the introduction section, SALL2 is the tumor suppressor gene but the IHC staining in human breast cancer shows very high SALL2 expression in ER+ breast cancer tissues (Fig. 2A). Authors have to discuss this observation. Further, they have to use the normal control breast tissues and compare SALL2 expression among normal control, primary breast cancer, and relapsed tamoxifen resistant breast cancer.

2) As shown in Fig. 3A-C, SALL2 is the upstream regulator of ESR1 and SALL2 silencing reduced ESR1 and ER α expression. Although SALL2 functions as a tumor suppressor and its expression in breast cancer is supposed to be very low to undetectable but ESR1 and ER α expression is

significantly upregulated in most of the ER-positive breast cancer. Further, SALL2 silencing reduced E2-induced tumor growth (Fig. 4A-B) and unresponsive to TAM treatment. Authors need to explain these contradictory observations.

3) As shown in Fig. 4D, SALL2 silenced MCF7 cells are able to form mammary tumor in the absence of E2 implantation. This suggests that tumor derived from SALL2 silencing is E2-independent. This is very interesting and breakthrough observation. Because, the scientific community always use estrogen pellet to induce MCF7 driven mouse xenograft. From now on they can use just SALL2 silenced MCF7 to induce xenograft in nude mouse. However, the authors did not show whether the tumor formed in SALL2 silenced MCF7 xenograft without E2 (Fig. 4D) is ER-positive or ER-negative. The authors need to stain the tumor tissue section with ER α antibody (as shown in Fig. 4C).

4) The authors need to show the SALL2 expression in normal immortalized mammary epithelial cell lines MCF10A and MCF12A, ER-positive, and ER-negative breast cancer cell lines. Also, in Tamoxifen sensitive and Tamoxifen resistant MCF7 cell lines.

5) Still lot of issues in the text of the manuscript, especially in the abstract section, and authors have to provide more attention to correct these issues.

Referee #2 (Remarks for Author):

This reviewer appreciates the effort of the authors to improve their manuscript with additional data, discussion, and appropriate interpretations. In general, this reviewer's comments and critiques have been adequately addressed.

Referee #3 (Comments on Novelty/Model System for Author):

Authors have used relevant model system. MCF-7 cell-derived model is the best for studies related to tamoxifen resistance

Referee #3 (Remarks for Author):

Authors have addressed concerns raised in the last review

2nd Revision - authors' response

19 September 2019

Referee #1 (Remarks for Author):

In this revised manuscript, the authors have successfully addressed most of the concerns raised by all the reviewers. They have performed several new experiments and provided very logical explanation for most of the critics raised by the reviewers. However, still some concerns or issues in the manuscripts:

1. As mentioned in the introduction section, SALL2 is the tumor suppressor gene but the IHC staining in human breast cancer shows very high SALL2 expression in ER+ breast cancer tissues (Fig. 2A). Authors have to discuss this observation. Further, they have to use the normal control breast tissues and compare SALL2 expression among normal control, primary breast cancer, and relapsed tamoxifen resistant breast cancer.

Response: We are thankful for the reviewer's comment and reviewer raised an important question. Transcription factor SALL2 has been reported to play vital roles in development and disease (1-2). Recently, SALL2 was found to be downregulated in several human cancer types, including breast cancer, ovarian cancer, and esophageal squamous cell carcinoma (3-5), and loss of SALL2 could abrogate serum deprivation-induced cell cycle arrest but SALL2 overexpression suppressed cell growth (3), suggesting that SALL2 acted as a tumor suppressor. However, Suvà et al found that SALL2 may also function as an oncogenic protein that played vital role in converting differentiated glioblastoma cells into cancer stem-like cells, resulting in glioblastom propagation (6). These results suggested that the functional roles of SALL2 may rely on cell types and cellular context. The abovementioned descriptions have been added in the Introduction section in the revised manuscript.

Highly proliferative solid tumor cells normally create regional hypoxia, lactic acid accumulation and nutrient deficiencies (7-8). Interestingly, it has reported that serum deprivation could induce SALL2 expression, which elicited cell cycle arrest (3). As expected, SALL2 expression was markedly expressed in tamoxifen-sensitive ER+ breast cancer tissues (Fig 2A). Concordantly, the human-protein-atlas also showed that multiple breast cancer tissues, such as Patient id: 2898, 1838 and 2369, displayed evidently detectable SALL2 level

(<https://www.proteinatlas.org/ENSG00000165821-SALL2/pathology/breast+cancer#img>).

However, promoter methylation resulted in SALL2 decreased in tamoxifen-resistant ER+ breast cancer tissues and ER- breast cancer tissues (Fig 2A and Fig 6A-C). Furthermore, we found that overexpression of SALL2 induced G1/S-arrest via upregulation of PTEN in breast cancer cells (Fig EV3B and Fig 5B and C). These results further supported the notion that SALL2 acts as a tumor suppressor in breast cancer. The abovementioned results and descriptions have been added in Discussion section in the revised manuscript.

As requested by the reviewer, we further examined SALL2 expression in the normal breast, primary breast cancer, and the relapsed tamoxifen resistant breast cancer tissues using IHC assay. Consistent with abovementioned mechanism, SALL2 expression was detectable in normal breast tissues but lower than that in primary tamoxifen-sensitive ER+ breast cancer tissues, and the relapsed tamoxifen resistant breast cancer tissues showed further decreased SALL2 (Appendix Fig S3B). These results have been incorporated into the revised manuscript.

References

- (1) de Celis JF, et al. Regulation and function of Spalt proteins during animal development. *Int J Dev Biol.* 2009; 53(8-10):1385-98.
- (2) Kelberman D, et al. Mutation of SALL2 causes recessive ocular coloboma in humans and mice. *Hum Mol Genet.* 2014; 23(10):2511-26.
- (3) Liu H, et al. A transcriptional program mediating entry into cellular quiescence. *PLoS Genet.* 2007; 3(6):e91.
- (4) Sung CK, et al. Promoter methylation of the SALL2 tumor suppressor gene in ovarian cancers. *Mol Oncol.* 2013; 7(3):419-27.
- (5) Luo J, et al. mRNA and methylation profiling of radioresistant esophageal cancer cells: the involvement of Sall2 in acquired aggressive phenotypes. *J Cancer.* 2017; 8(4): 646-656.
- (6) Suvà ML, et al. Reconstructing and reprogramming the tumor-propagating potential of glioblastoma stem-like cells. *Cell.* 2014; 157(3):580-94.
- (7) Hanahan D, et al. Hallmarks of cancer: the next generation. *Cell.* 2011; 144(5):646-74.
- (8) Katheder NS, et al. Microenvironmental autophagy promotes tumour growth. *Nature.* 2017; 541(7637):417-420.

2. As shown in Fig. 3A-C, SALL2 is the upstream regulator of ESR1 and SALL2 silencing reduced ESR1 and ER expression. Although SALL2 functions as a tumor suppressor and its expression in breast cancer is supposed to be very low to undetectable but ESR1 and ER α expression is significantly upregulated in most of the ER-positive breast cancer. Further, SALL2 silencing reduced E2-induced tumor growth (Fig. 4A-B) and unresponsive to TAM treatment. Authors need to explain these contradictory observations.

Response: We do appreciate the reviewer for raising this important point. Transcription factor SALL2 has been reported to play vital roles in development and disease (1-2). Recently, SALL2 was found to be downregulated in several human cancer types, including breast cancer, ovarian cancer, and esophageal squamous cell carcinoma (3-5), and loss of SALL2 could abrogate serum deprivation-induced cell cycle arrest but SALL2 overexpression suppressed cell growth (3), suggesting that SALL2 acted as a tumor suppressor. However, Suvà et al found that SALL2 may also function as an oncogenic protein that played vital role in converting differentiated glioblastoma cells into cancer stem-like cells, resulting in glioblastoma propagation (6). These results suggested that the functional roles of SALL2 may rely on cell types and cellular context. In the current study, we demonstrated that SALL2 could simultaneously upregulate ER α and PTEN through directly binding to their promoters, suggesting that SALL2 may play dual function in breast cancer cells. Meanwhile, it has previously reported that serum deprivation could induce SALL2 expression, which elicited cell cycle arrest (3). Since highly proliferative solid tumor cells normally create regional hypoxia and nutrient deficiencies (7-8), which implicated that SALL2 levels might be induced in breast cancer tissues. As expected, SALL2 expression was markedly expressed and positively correlated ER α expression in ER+ breast cancer tissues, which resulted in tamoxifen sensitive (Fig 2A and Fig 4F-I). However, promoter methylation-mediated SALL2 reduction in ER+ breast cancer tissues resulted in downregulation of ER α and PTEN, which led to reduced E2-

induced tumor growth and unresponsive to TAM treatment (Fig 3-Fig 6). Importantly, *in vivo* experiments showed that DNA methyltransferase inhibitor-mediated SALL2 restoration resensitized tamoxifen-resistant breast cancer to tamoxifen therapy (Fig 7). Therefore, our results shed light on the mechanism of SALL2 in regulation of ER and represent a potential clinical signature that used for subgrouping breast cancer patients who might be benefit from co-therapy with tamoxifen and DNMTs inhibitor. The abovementioned descriptions have been added in the Introduction section in the revised manuscript.

References

- (1) de Celis JF, et al. Regulation and function of Spalt proteins during animal development. *Int J Dev Biol.* 2009; 53(8-10):1385-98.
- (2) Kelberman D, et al. Mutation of SALL2 causes recessive ocular coloboma in humans and mice. *Hum Mol Genet.* 2014; 23(10):2511-26.
- (3) Liu H, et al. A transcriptional program mediating entry into cellular quiescence. *PLoS Genet.* 2007; 3(6):e91.
- (4) Sung CK, et al. Promoter methylation of the SALL2 tumor suppressor gene in ovarian cancers. *Mol Oncol.* 2013; 7(3):419-27.
- (5) Luo J, et al. mRNA and methylation profiling of radioresistant esophageal cancer cells: the involvement of Sall2 in acquired aggressive phenotypes. *J Cancer.* 2017; 8(4): 646-656.
- (6) Suvà ML, et al. Reconstructing and reprogramming the tumor-propagating potential of glioblastoma stem-like cells. *Cell.* 2014; 157(3):580-94.
- (7) Hanahan D, et al. Hallmarks of cancer: the next generation. *Cell.* 2011; 144(5):646-74.
- (8) Katheder NS, et al. Microenvironmental autophagy promotes tumour growth. *Nature.* 2017; 541(7637):417-420.

3. As shown in Fig. 4D, SALL2 silenced MCF7 cells are able to form mammary tumor in the absence of E2 implantation. This suggests that tumor derived from SALL2 silencing is E2-independent. This is very interesting and breakthrough observation. Because, the scientific community always use estrogen pellet to induce MCF7 driven mouse xenograft. From now on they can use just SALL2 silenced MCF7 to induce xenograft in nude mouse. However, the authors did not show whether the tumor formed in SALL2 silenced MCF7 xenograft without E2 (Fig. 4D) is ER-positive or ER-negative. The authors need to stain the tumor tissue section with ER antibody (as shown in Fig. 4C).

Response: We thank for the reviewer's comment and the reviewer's point is well taken. As requested by the reviewer, the ER α expression was further examined in SALL2 silenced MCF7 xenograft without E2 treatment. As shown in new Fig 4E, ER α expression was barely detected in SALL2-silenced tumors without E2 but strongly expressed in E2-treated MCF7/control xenograft, which further support the notion that silencing SALL2 confers estrogen-independent tumorigenicity of ER+ breast cancer cells. The abovementioned results have been incorporated into Fig 4E in the revised manuscript.

4. The authors need to show the SALL2 expression in normal immortalized mammary epithelial cell lines MCF10A and MCF12A, ER-positive, and ER-negative breast cancer cell lines. Also, in Tamoxifen sensitive and Tamoxifen resistant MCF7 cell lines.

Response: The reviewer's point is well taken. As requested by the reviewer, the SALL2 expression was further examined in the cell lines as indicated by the reviewer. As shown in Appendix Fig S3C, the expression of SALL2 in ER-positive breast cancer cell lines, including MCF7, T47D, ZR-75-1, MDA-MB-361, MDA-MB-415 and ZR-75-30, was significantly higher than that in ER-negative breast cancer cell lines (BT-20, BT-549 and MDA-MB-231) and in normal immortalized mammary epithelial cell lines MCF10A and MCF12A, which are also ER-negative cell lines. Consistently, SALL2 levels were significantly decreased in Tamoxifen resistant MCF7 cell line compared to Tamoxifen sensitive MCF7 cell line (Fig EV2B and C). These results further support the notion that SALL2 contributes to regulation of ER. The abovementioned results has been incorporated into Appendix Fig S3C in the revised manuscript.

5. Still lot of issues in the text of the manuscript, especially in the abstract section, and authors have to provide more attention to correct these issues.

Response: We are thankful for the reviewer's comment and sorry for the grammatical and spelling mistakes in the last submitted manuscript. We have carefully edited the entire manuscript and had the revised manuscript corrected again by professional editors before this resubmission.

Referee #2 (Remarks for Author):

This reviewer appreciates the effort of the authors to improve their manuscript with additional data, discussion, and appropriate interpretations. In general, this reviewer's comments and critiques have been adequately addressed.

Response: We thank the reviewer for the positive comment on our revised manuscript.

Referee #3 (Remarks for Author):

Authors have addressed concerns raised in the last review

Response: We deeply thank the reviewer for his (her) appreciation on our tremendous efforts in addressing all the concerns.